# Structural features of heteromeric channels composed of CALHM2 and CALHM4 paralogs

Katarzyna Drożdżyk, Martina Peter, Raimund Dutzler*

Department of Biochemistry, University of Zurich, Zurich, Switzerland

**Abstract** The CALHM proteins constitute a family of large pore channels that contains six closely related paralogs in humans. Two family members, CALHM1 and 3, have been associated with the release of ATP during taste sensation. Both proteins form heteromeric channels that activate at positive potential and decreased extracellular $Ca^{2+}$ concentration. Although the structures of several family members displayed large oligomeric organizations of different size, their function has in most cases remained elusive. Our previous study has identified the paralogs CALHM2, 4 and, 6 to be highly expressed in the placenta and defined their structural properties as membrane proteins exhibiting features of large pore channels with unknown activation properties (Drożdżyk et al., 2020). Here, we investigated whether these placental paralogs would form heteromers and characterized heteromeric complexes consisting of CALHM2 and CALHM4 subunits using specific binders as fiducial markers. Both proteins assemble with different stoichiometries with the largest population containing CALHM2 as the predominant component. In these oligomers, the subunits segregate and reside in their preferred conformation found in homomeric channels. Our study has thus revealed the properties that govern the formation of CALHM heteromers in a process of potential relevance in a cellular context.

*For correspondence:
dutzler@bioc.uzh.ch

**Competing interest:** The authors declare that no competing interests exist.

## eLife assessment

In this interesting study, Drożdżyk and colleagues analyze the ability of placental CALHM orthologs to form stable complexes, identifying that CALHM2 and CALHM4 form heterooligomeric channels. The authors then determine cryo-EM structures of heterooligomeric CALHM2 and CALHM4 that reveal a distinct arrangement in which the two orthologs can interact, but preferentially segregate in the channel. This is an **important** study; the data provide **compelling** support for the interpretations and overall, the work is clearly described.

## Introduction

Large pore channels encompass a heterogeneous group of membrane proteins in higher eukaryotes, which form large oligomeric assemblies that facilitate the diffusion of diverse substrates with poor selectivity, including metabolites and signaling molecules such as the nucleotide ATP (*Syrjanen et al., 2021*). The calcium homeostasis modulators (CALHM) constitute one family of large pore channels that contains six members in humans (*Ma et al., 2016*). Their role in physiology is currently best understood for the paralogs CALHM1 and 3. Both proteins are expressed in type II taste bud cells to contribute to the sensation of umami, sweet, and bitter taste (*Ma et al., 2018b*; *Taruno et al., 2013b*). There, they are involved in the non-vesicular release of ATP as a consequence of membrane depolarization following the initial sensation of tastants by G-protein coupled receptors (*Taruno et al., 2013a*). ATP in turn acts as a neurotransmitter to activate P2X-receptors on the postsynaptic side.

CALHM proteins reside in a non-conducting state at resting potentials and are activated in response to membrane depolarization with a decrease in extracellular $Ca^{2+}$ shifting the threshold of activation towards more hyperpolarizing voltages, thereby allowing conduction also at negative potentials (*Ma et al., 2012*). The open channel shows poor selectivity and is permeable to the multivalent anion ATP. Although these properties are shared by homomeric assemblies of CALHM1 in heterologous expression systems, their slow kinetics of activation would prohibit fast signaling and thus does not reflect the behavior observed in taste bud cells (*Ma et al., 2012*). To enhance its activation properties, CALHM1 assembles as heteromeric complex with its paralog CALHM3, which on its own does not form functional ion channels (*Ma et al., 2018b*). The physiological role of other CALHM paralogs is currently much less well understood, although the ubiquitously expressed CALHM2 was also associated with ATP release (*Liao et al., 2023*; *Ma et al., 2018a*) and CALHM6 was recently described as channel in immunological synapses of macrophages (*Danielli et al., 2023*).

Based on their assumed topology containing four membrane-spanning segments, the CALHM family was originally predicted to share a close structural relationship with large pore channels of the connexin, pannexin, and LRRC8 families (*Siebert et al., 2013*). However, this relationship was refuted in numerous structures of homomeric assemblies of different CALHM paralogs that have recently been determined by cryo-electron microscopy (cryo-EM) from samples obtained from heterologous expression in mammalian and insect cell cultures (*Choi et al., 2019*; *Demura et al., 2020*; *Drożdżyk et al., 2020*; *Foskett, 2020*; *Liu et al., 2020*; *Ren et al., 2022*; *Ren et al., 2020*; *Syrjanen et al., 2020*; *Yang et al., 2020*). Although sharing the same number of membrane-spanning segments, their arrangement is unique and differs from other large pore channels, with an extended helix located on the C-terminus and forming the core of a cytoplasmic domain further mediating subunit interactions. These structures show channels that assemble into higher oligomers around an axis of symmetry that presumably defines the conduction path. These assemblies are heterogeneous in nature, with paralogs frequently adopting different oligomeric states in the same sample. The smallest assemblies were generally observed for CALHM1, where structures from different species contained between seven and nine subunits (*Demura et al., 2020*; *Ren et al., 2022*; *Ren et al., 2020*; *Syrjänen et al., 2023*; *Syrjanen et al., 2020*). The size of the respective assemblies is generally larger for other CALHM paralogs, ranging from decamers to tridecamers (*Choi et al., 2019*; *Drożdżyk et al., 2020*; *Liu et al., 2020*; *Syrjanen et al., 2020*). In detergent solution, these channels do in certain cases pair to form gap-junction-like organizations, although there is currently no evidence for such transcellular interactions in a physiological environment (*Choi et al., 2019*; *Syrjanen et al., 2020*). In oligomeric channels, the pore is lined by the first transmembrane segment (TM1) of each subunit, which forms an inner, of two concentric rings, with the outer ring being constituted by TM2-4, with TM2 and 4 of neighboring subunits engaging in extended interactions (*Choi et al., 2019*; *Drożdżyk et al., 2020*; *Liu et al., 2020*; *Syrjanen et al., 2020*). The loosely packed TM1 is found in distinct conformations in different structures to shape the pore geometry and thus presumably alter conduction properties (*Choi et al., 2019*; *Drożdżyk et al., 2020*).

We have previously studied the structural and functional properties of the CALHM paralogs 2, 4 and 6, whose transcripts were found to be abundant in placental epithelia in dependence on the developmental stage of the organ (*Drożdżyk et al., 2020*). These placental paralogs all form large oligomers, which in CALHM4 consist of similar-sized populations of decameric and undecameric assemblies in the same sample. Both oligomers were found to pair via contacts on their intracellular side into large tubular structures. In this sample, both assemblies adopt equivalent subunit conformations, leading to large cylindrical pores that are constricted by a short N-terminal helix preceding TM1, which is oriented parallel to the membrane plane and projects towards the pore axis (*Drożdżyk et al., 2020*). In the same study, a similar pairing of channels containing between 11 and 12 subunits was also found for CALHM2, where the preferred orientation of the particles within the sample has prevented a three-dimensional reconstruction at high-resolution. In contrast, no dimerization was detected in a CALHM6 preparation, where, although also existing as undecamers, a decameric assembly was prevalent and thus better resolved. As a consequence of a conformational rearrangement of TM1, which has detached from its interaction with the rest of the subunit, resulting in a higher mobility, the CALHM6 pore has adopted a conical shape with a decreasing diameter towards the intracellular side suggesting that it might reside in a different functional state (*Drożdżyk et al., 2020*).

Despite the observed large assemblies, none of the three paralogs overexpressed in *X. laevis* oocytes showed any activity in electrophysiology experiments under conditions where CALHM1 was conductive, regardless of their expression at the cell surface, suggesting that they may be activated by different stimuli (*Drożdżyk et al., 2020*). This was unexpected in light of the distinct conformations observed for CALHM4 and 6 and their large apparent pore diameter, which is particularly pronounced in case of CALHM4. In this respect, it was remarkable to find continuous density with bilayer-like features inside the pore of CALHM4, matching the hydrophobic character of the pore lining, which might explain the lack of activity observed for this protein (*Drożdżyk et al., 2020*). This density is not found in CALHM6, where the altered shape of the pore could not accommodate a bilayer. The functional properties of placental CALHM paralogs thus remained elusive.

Since the formation of heteromers composed of CALHM1 and 3 paralogs strongly altered the functional behavior of the respective homomers, we wondered whether the tree paralogs expressed in the placenta would share a similar property and whether their heteromerization would increase their activity under the investigated conditions. To clarify this question, we have here investigated the ability of the overexpressed CALHM homologs 2, 4, and 6, to form heteromeric assemblies. Robust interactions were identified between CALHM2 and 4, but to a much lower extent between any of the two paralogs and CALHM6. The subsequent structural characterization of CALHM2/4 heteromers, employing synthetic subunit-specific nanobodies (sybodies), showed channels with a broad subunit distribution. The largest populations contain a predominance of CALHM2 with two to four CALHM4 subunits, where both paralogs segregate in the heteromeric assemblies. While subunits maintain their distinct structural features found in homomeric channels, they also influence the conformations of contacting subunits.

## Results

### Formation of CALHM heteromers

The formation of heteromers is a hallmark of different eukaryotic ion channels, with their subunit composition shaping functional properties. Such heteromerization has also been found in the CALHM family of large-pore channels, where the paralogs CALHM1 and 3 oligomerize to form channels that are activated by membrane depolarization and the decrease of the extracellular $Ca^{2+}$-concentration (*Ma et al., 2018b*). To probe whether this property would also extend to other family members, we have initially investigated the ability of the three placental paralogs CALHM2, 4, and 6 to form heteromeric assemblies. To this end, we have co-transfected HEK293S cells with pairs of constructs coding for different paralogs and containing unique tags for affinity purification. After solubilization, specific subunits were captured via their attached tags and the sample was assayed for the co-purification of the second subunit by SDS-PAGE and Western blot. In that way, we have identified robust heteromerization of the paralogs CALHM2 and 4, and weak pairing of both paralogs with CALHM6, thus suggesting that specific heteromerization might be a property that is inherent to certain CALHM paralogs (*Figure 1A*).

After identifying the heteromerization of CALHM2 and 4, we have tested whether this process has altered the activation properties of these channels. To this end, we have expressed either single or pairs of different CALHM paralogs in HEK293 cells and investigated their functional properties by patch-clamp electrophysiology. When recorded in the whole-cell configuration, we were able to reproduce the previously observed phenotype of CALHM1 to form channels that activate at positive voltages (*Figure 1B*, *Figure 1—figure supplement 1A*). The observed functional properties are enhanced upon co-expression of CALHM1 and 3 subunits, which results in the increase of currents and activation kinetics compared to CALHM1, when recorded in the same manner (*Figure 1C*, *Figure 1—figure supplement 1B*). In contrast, neither CALHM2 nor CALHM4 shows appreciable currents under equivalent conditions, which conforms with a previous investigation performed in *X. laevis* oocytes (*Drożdżyk et al., 2020*; *Figure 1D*, *Figure 1—figure supplement 1C*). Finally, we have investigated whether their assembly as heteromers would enhance the activity of CALHM2 and CALHM4 and thus proceeded with the functional characterization of cells that were co-transfected with plasmids encoding each subunit. However, similar to the respective homomers, we were unable to detect any current response at a positive voltage, thus emphasizing that these proteins share an activation mechanism that is distinct from CALHM1/3 (*Figure 1E and F*, *Figure 1—figure supplement 1D, E*).

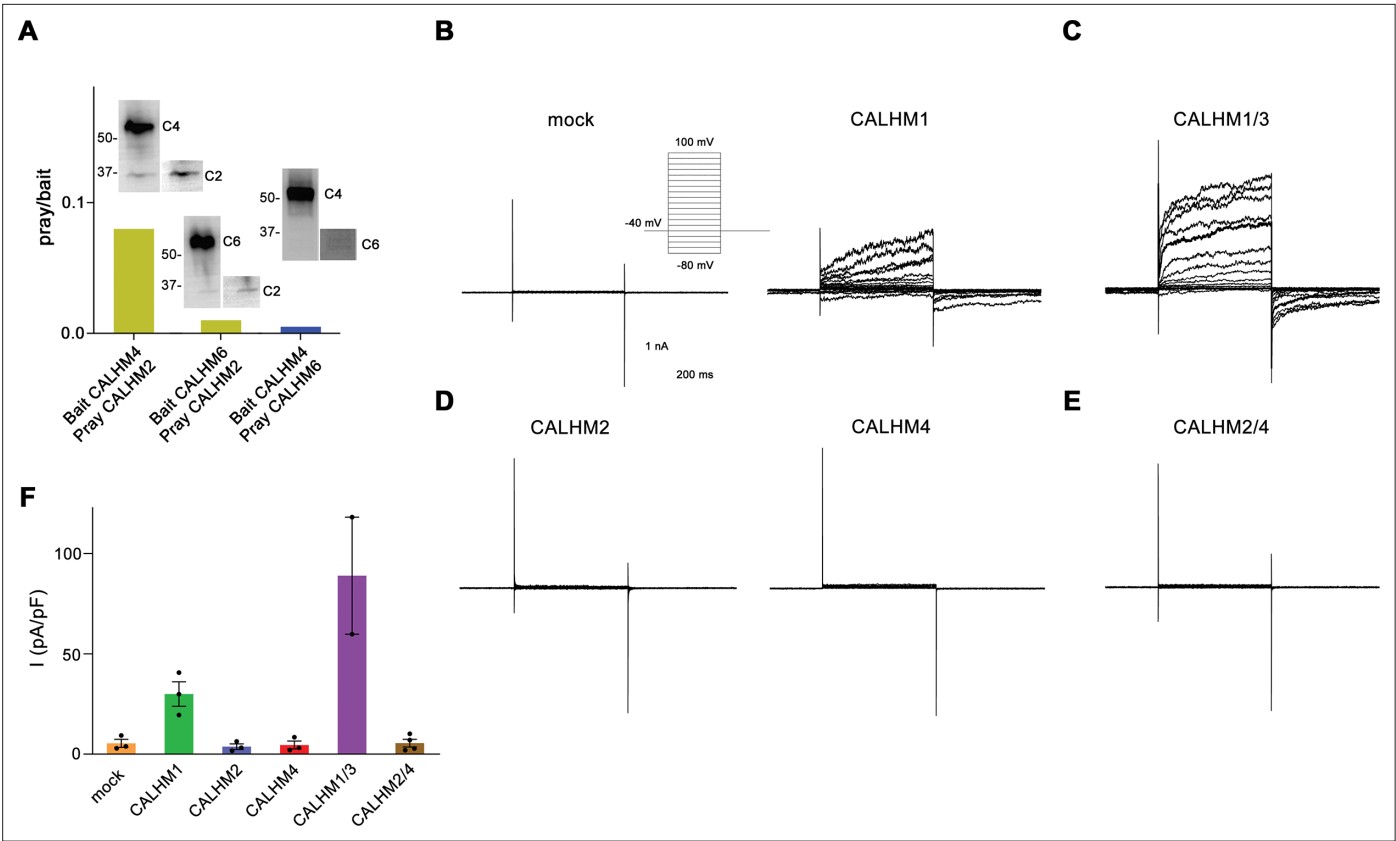

**Figure 1.** Heteromerization and functional properties of placental calcium homeostasis modulators (CALHM) paralogs. (**A**) Western blot (detecting the myc-tag attached to the C-termini of respective constructs) of CALHM subunits expressed upon co-transfection of HEK293S GnTI⁻ cells with pairs of CALHM subunits. CALHM channels were isolated by affinity purification of one subunit containing a fusion to Venus and an SBP-tag (bait, shown at higher molecular weight). The resulting samples contain a mix of homomers of the purified subunit and heteromers (with the second subunit not containing Venus and SBP tags, prey at lower molecular weight). The ratio of pray/bait is displayed based on the integration of the intensity of the displayed Western blots. The experiment has been carried out once. Molecular weights are indicated. (**B–E**) Representative patch-clamp electrophysiology recordings (whole-cell configuration) of indicated CALHM subunits expressed in HEK-293 cells measured in buffers containing 2 mM Ca²⁺ on the extracellular side. (**B**) Comparison of currents from mock-transfected cells (left) and cells expressing CALHM1 (right). The inset shows the voltage protocol. (**C**) Current response from cells co-transfected with DNA coding for CALHM1 and CALHM3 subunits. (**D**) Current response from cells transfected with CALHM2 (left) or CALHM4 (right) subunits. (**E**) Current response from cells co-transfected with CALHM2 and CALHM4 subunits. (**F**) Mean current density of recordings of different CALHM constructs (measured at 100 mV, 400 ms after the voltage step), errors are SEM for n>2 and differences from the mean in case of n=2, values from individual recordings are shown as circles.

The online version of this article includes the following source data and figure supplement(s) for figure 1:

**Source data 1.** Uncropped images of immunoblots and raw data plotted in *Figure 1A*.

**Source data 2.** Raw recordings displayed and plotted in *Figure 1B–F*.

**Figure supplement 1.** Electrophysiology data of calcium homeostasis modulators (CALHM) paralogs recorded at low extracellular Ca²⁺.

**Figure supplement 1—source data 1.** Raw recordings displayed and plotted in *Figure 1—figure supplement 1*.

## Selection of paralog-specific binders

Following, we set out to characterize the structural properties of CALHM2/4 heteromers by cryo-EM. This task is complicated by the unknown arrangement of subunits sharing similar structural features and the potentially large compositional heterogeneity of heteromers, which might prohibit the classification and alignment of oligomeric complexes. We have thus initially engaged in the generation of paralog-specific binders that can be used as fiducial markers to aid the identification of subunits in heteromeric channels. As in previous projects, we have decided for synthetic nanobodies (termed sybodies), which were selected from large construct libraries by different display methods (***Zimmermann et al., 2020***). In independent efforts, we have isolated binders against homomeric assemblies of CALHM2 and CALHM4 (***Figure 2—figure supplement 1***) and subsequently used them for a structural

characterization of respective homomers to define their binding epitopes and their effect on subunit conformations.

## Features of the CALHM4/SbC4 complex

Our selection against CALHM4 was carried out with paired assemblies of channels containing either ten or eleven subunits each, which is reflected in the low elution volume of the sample during size exclusion chromatography (*Figure 2—figure supplement 1B*). Dimerization in CALHM4 proceeds at its intracellular region, resulting in large particles, which expose their extracellular parts at both ends of a tubular assembly (*Drożdżyk et al., 2020*). This sample has allowed us to isolate a specific binder, termed Sb^CALHM4 or short SbC4, that forms a tight complex with the CALHM4 channel pair, which remains intact during size exclusion chromatography as confirmed by SDS-PAGE (*Figure 2—figure supplement 1A*, B). Upon characterization by surface plasmon resonance, we measured a binding affinity of about 100 nM (*Figure 2—figure supplement 1C*). Conversely, we did not find any evidence of this protein to engage in strong interactions with CALHM2 (*Figure 2—figure supplement 1D*). When investigated by cryo-EM, we observed two populations of particles either exhibiting D10 or D11 symmetry (corresponding to dimers of decameric and undecameric channels) that closely resemble equivalent structures of CALHM4 determined in previous studies (*Drożdżyk et al., 2020*; *Figure 2—figure supplement 2*). Between the two populations, the reconstruction of the density with D10 symmetry imposed was of higher resolution and we thus restrict our detailed analysis to this structure (*Figure 2A and B*, *Figure 2—figure supplement 2*, *Table 1*). Subunits in the oligomeric assembly are arranged around an axis of symmetry, which presumably defines the pore of a large channel (*Figure 2B*). Each subunit contains four membrane-spanning helices, three of which (TM2-4) are tightly interacting to form an outer ring with a large diameter that separates the inside of the pore from its surrounding membrane environment (*Figure 2C and D*). In this ring, TM2 and 4 of adjacent subunits engage in extended interactions (*Figure 2D*). In contrast, TM1 forms an inner ring of helices lining the pore, with the bent helical N-terminus (NH) running perpendicular to the membrane and projecting towards the pore axis (*Figure 2D*). TM1 packs against TM3 of the outer ring but is not involved in direct subunit interactions. Within each channel, we find residual density of low resolution, whose distribution resembles the headgroup region of a lipid bilayer, which was thus previously ascribed to a membrane assembled in the lumen of the pore (*Drożdżyk et al., 2020*; *Figure 2E*). In this cryo-EM map, we find the density of sybodies binding at the extracellular side of each subunit forming two rings of binders at both ends of the assembly (*Figure 2A and B*). Although not sufficiently well resolved to define detailed interactions with certainty, neighboring sybodies appear not to clash, with epitopes being confined to a single subunit, which provides ideal properties for the identification of CALHM4 subunits in heteromeric channels.

## Features of the CALHM2/SbC2 complex

In case of CALHM2, our selection efforts have allowed us to identify a binder termed Sb^CALHM2 (short SbC2) forming tight complexes with homomeric assemblies (*Figure 2—figure supplement 1A, E–G*). This sybody remains bound to the channel during size exclusion chromatography, which is manifested in the decreased elution volume of the complex and the presence of both proteins in peak fractions as confirmed by SDS-PAGE (*Figure 2—figure supplement 1E*). As in case of SbC4, the binding is subunit specific (*Figure 2—figure supplement 1F, G*). We have then proceeded with the structural characterization of CALHM2/SbC2 complexes by cryo-EM and found two populations of particles showing either C10 or C11 symmetry. In this dataset, only the reconstruction of undecameric particles resulted in high-resolution density that resembles structures of CALHM2 determined in previous studies (*Choi et al., 2019*; *Demura et al., 2020*; *Syrjanen et al., 2020*; *Figure 3A and B*, *Figure 3—figure supplement 1*, *Table 1*). In contrast to the CALHM4 structure, we did not detect any dimerization of channels (*Figure 3—figure supplement 1*). Another pronounced difference in the CALHM2 structure concerns the conformation of TM1, which is detached from its interaction with TM3 as found in CALHM4 and instead projects towards the pore axis (*Figure 3C*). Due to its high mobility, the density of this helix is only defined at its C-terminal end at the boundary to the extracellular loop connecting to TM2, whereas its remainder towards the N-terminus is not resolved (*Figure 3C*). This arrangement of TM1 in an 'up' position resembles the conformation observed in a previous structure of a complex of CALHM2 with its putative inhibitor rubidium red, although its density in the inhibitor complex is better defined

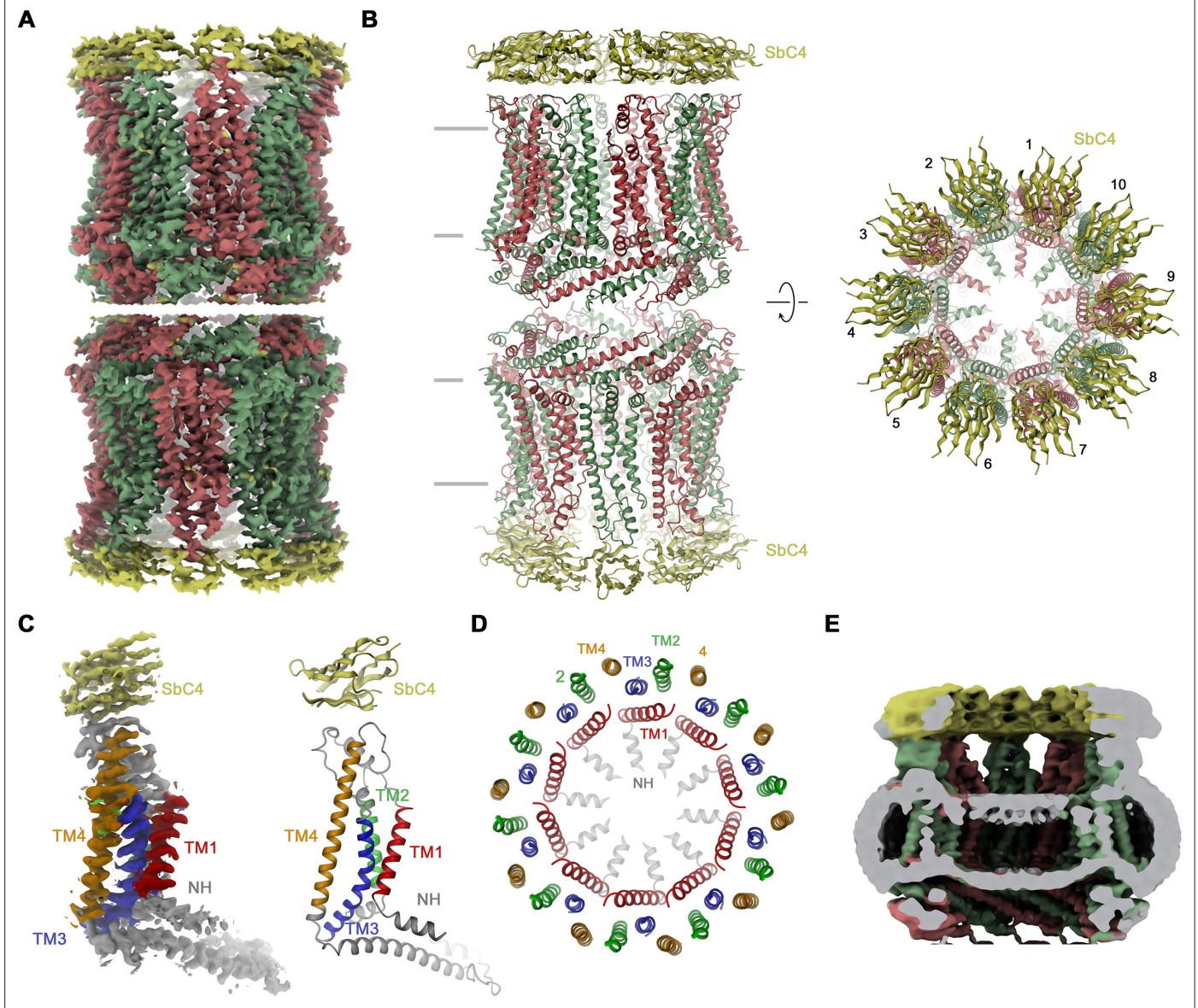

**Figure 2.** Structural characterization of the CALHM4/SbC4 complex. (**A**) Cryo-electron microscopy (Cryo-EM) density at 3.7 Å (after application of D10 symmetry) of a pair of decameric CALHM4/SbC4 complexes interacting via their cytoplasmic regions. (**B**) Ribbon representation of the same complex viewed parallel (left) and perpendicular (right) to the membrane plane. Membrane boundaries are indicated (left). (**A, B**), CALHM4 subunits are colored in red and green, SbC4 in yellow. (**C**) Cryo-EM density (left) and ribbon representation (right) of a single subunit of the CALHM4/SbC4 complex. Transmembrane helices are labeled and shown in unique colors. (**D**) Organization of transmembrane (TM) helices in the decameric channel. The view is from the extracellular side, the coloring is as in **C**. (**E**) Slab through a low-pass filtered map of the CALHM4/SbC4 complex (viewed from within the membrane) shows bilayer-like density in the lumen of the wide pore.

The online version of this article includes the following source data and figure supplement(s) for figure 2:

**Figure supplement 1.** Sybody selection and characterization.

**Figure supplement 1—source data 1.** SDS-PAGE gel images and size-exclusion chromatography data plotted in *Figure 2—figure supplement 1B, E*.

**Figure supplement 1—source data 2.** Surface plasmon resonance (SPR) binding data plotted in *Figure 2—figure supplement 1C, D, F, G*.

**Figure supplement 2.** Cryo-electron microscopy (Cryo-EM) reconstruction of CALHM4 in complex with sybody SbC4.

(*Choi et al., 2019*). In contrast to CALHM4, where TM3 and TM4 of the same subunit tightly interact throughout, in CALHM2 both helices have moved apart at their intracellular halve (*Figure 3D*). In the structure, the resulting narrow fenestration connecting the channel interior and the membrane is filled by a bound lipid (*Figure 3D*). Finally, unlike in the CALHM4 structure where TM1 resides in a 'down'

**Table 1.** Cryo-electron microscopy (Cryo-EM) Data collection, refinement, and validation statistics.

| | CALHM4/SbC4 | CALHM2/SbC2 | CALHM2/CALHM4/SbC4 | CALHM2/CALHM4/SbC2/SbC4 |
|---|---|---|---|---|
| | EMD-19365 PDB 8RMN | EMD-19362 PDB 8RMK | EMD-19363 PDB 8RML | EMD-19364 PDB 8RMM |
| **Data collection and processing** | | | | |
| Microscope | FEI Titan Krios G3i | FEI Titan Krios G3i | FEI Titan Krios G3i | FEI Titan Krios G3i |
| Camera | Gatan K3 +GIF | Gatan K3 +GIF | Gatan K3 +GIF | Gatan K3 +GIF |
| Magnification | 130,000 | 130,000 | 130,000 | 130,000 |
| Voltage (kV) | 300 | 300 | 300 | 300 |
| Electron exposure (e$^-$/Å$^2$) | 70 | 72 | 60 | 60 |
| Defocus range (μm) | –1.0 to –2.4 | –1.0 to –2.4 | –1.0 to –2.4 | –1.0 to –2.4 |
| Pixel size* (Å) | 0.651 (0.3255) | 0.651 (0.3255) | 0.651 (0.3255) | 0.651 (0.3255) |
| Initial particle images (no.) | 424,942 | 1,068,223 | 1,345,836 | 2,800,265 |
| Final particle images (no.) | 52,248 | 37,894 | 131,652 | 93,191 |
| Symmetry imposed | D10 | C11 | C1 | C1 |
| Map resolution (Å) | | | | |
| FSC threshold 0.143 | 3.7 | 3.07 | 3.84 | 3.26 |
| Map resolution range (Å) | 3.5–5 | 2.6–5 | 3.5–6 | 3–6 |
| **Refinement** | | | | |
| Model resolution (Å) | | | | |
| FSC threshold 0.5 | 3.9 | 3.2 | 4.1 | 3.7 |
| Map sharpening b-factor (Å$^2$) | –152.97 | –101.8 | –113.8 | –67.2 |
| Model composition | | | | |
| Non-hydrogen atoms | 55,460 | 36,113 | 24,456 | 32,891 |
| Protein residues | 7199 | 4521 | 3089 | 4163 |
| Ligand (PLC) | | 11 | 0 | 6 |
| *B* factors (Å$^2$) | | | | |
| Protein | 74.47 | 75.52 | 121.74 | 69.29 |
| Ligand | | 59.93 | | 80.08 |
| R.M.S. deviations | | | | |
| Bond lengths (Å) | 0.004 | 0.003 | 0.004 | 0.003 |
| Bond angles (°) | 0.710 | 0.611 | 0.596 | 0.671 |
| **Validation** | | | | |
| MolProbity score | 2.00 | 1.86 | 1.87 | 1.85 |
| Clashscore | 13.63 | 12.04 | 12.92 | 11.37 |
| Poor rotamers (%) | 1.16 | 0.45 | 0.46 | 0.79 |
| Ramachandran plot | | | | |
| Favored (%) | 95.60 | 96.05 | 96.28 | 96 |
| Allowed (%) | 4.40 | 3.95 | 3.72 | 3.98 |

*Table 1 continued on next page*

*Table 1 continued*

| | CALHM4/SbC4 | CALHM2/SbC2 | CALHM2/CALHM4/SbC4 | CALHM2/CALHM4/SbC2/SbC4 |
|---|---|---|---|---|
| | EMD-19365 PDB 8RMN | EMD-19362 PDB 8RMK | EMD-19363 PDB 8RML | EMD-19364 PDB 8RMM |
| Disallowed (%) | 0.00 | 0.00 | 0.00 | 0.02 |

*Values in parentheses indicate the pixel size in super-resolution.

conformation, the lumen of the CALHM2 channel does not contain pronounced bilayer-like residual density (*Figure 3E*). In the CALHM2 assembly, strong density of appropriate size and shape at each subunit defines the location of SbC2 binding to the intracellular side of the channel via its extended CDR3 loop (*Figure 3A–E*). On CALHM2, the epitope encompasses residues located on the intracellular end of TM3, the region immediately following TM4, and part of the C-terminal loop connecting the short helices CT2H and CT3H of the contacted subunit. Additionally, the very C-terminus of the adjacent subunit (CT$^{+1}$, located in a counterclockwise position when viewed from the extracellular side), which contains six additional residues added to the expression construct contributes to the

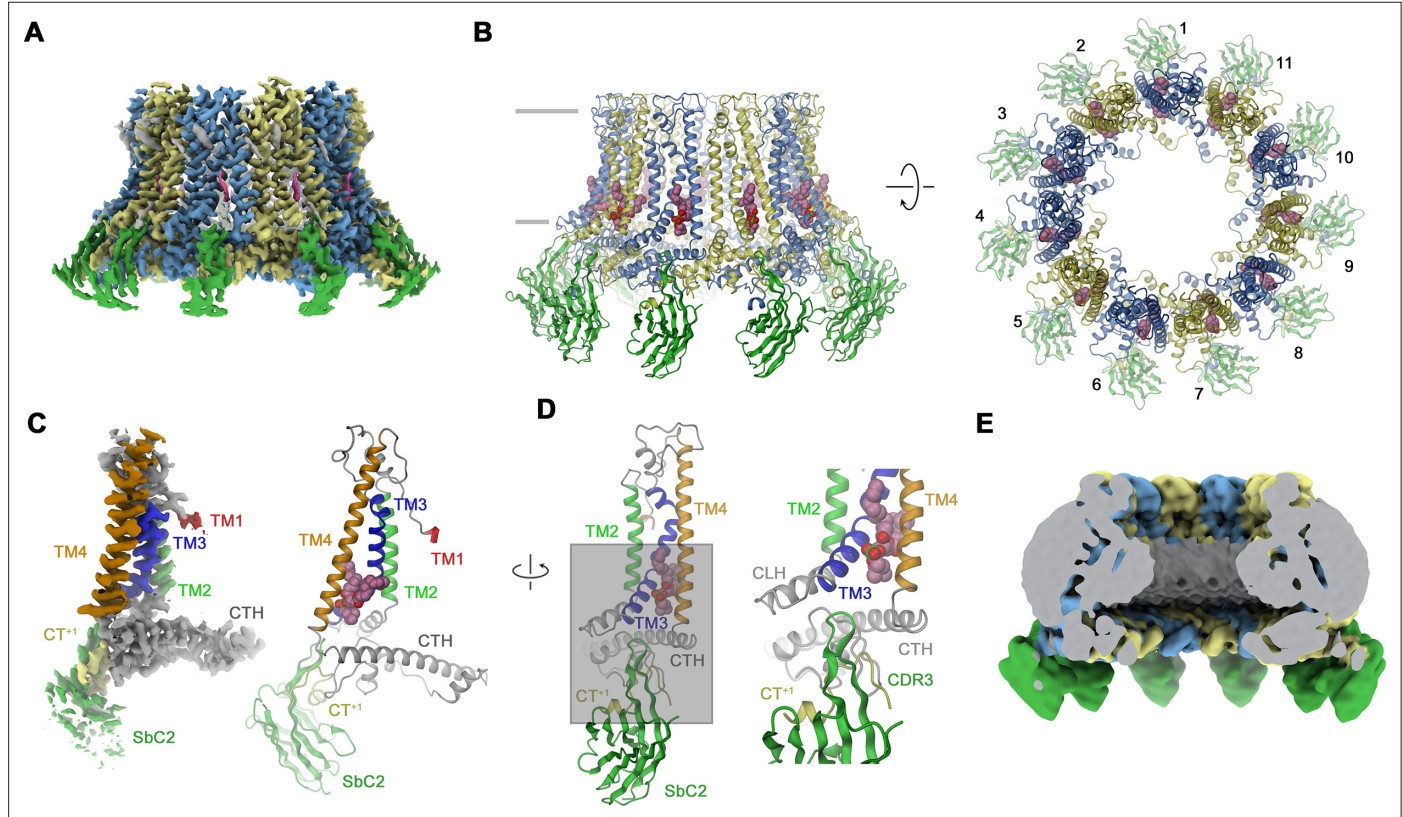

**Figure 3.** Structural characterization of a CALHM2/SbC2 complex. (**A**) Cryo-electron microscopy (Cryo-EM) density of the undecameric CALHM2/SbC2 complex at 3.1 Å (after application of C11 symmetry). (**B**) Ribbon representation of the same complex viewed parallel to the membrane plane (left) and from the extracellular side (right). Membrane boundaries are indicated (left). The space filling model shows a bound lipid. (**A, B**), CALHM2 subunits are colored in yellow and blue, SbC2 in green, and the bound lipid in magenta. (**C**) Cryo-EM density (left) and ribbon representation (right) of a single subunit of the CALHM2/SbC2 complex. Transmembrane helices are labeled and shown in unique colors. (**D**) Subunit (with orientation relative to C indicated) and blow-up of the sybody-interaction region. The C-terminus of the adjacent subunits contributing to the interaction epitope is colored in yellow and labeled (CT$^{+1}$). (**E**) Slab through a low-pass filtered map of the CALHM2/SbC2 complex (viewed from within the membrane) does not show bilayer-like density in the lumen of the pore.

The online version of this article includes the following figure supplement(s) for figure 3:

**Figure supplement 1.** Cryo-electron microscopy (Cryo-EM) reconstruction of CALHM2 in complex with sybody SbC2.

interaction (*Figure 3D*). This C-terminus is unstructured in the absence of the sybody but contacts the region preceding CT3H on the neighboring CALHM2 subunit and the CDR3 region of SbC2 to become partly buried in the binding interface (*Figure 3D*). As a consequence, about 75% of the binding interface covering about 2'200 Å$^2$ of the combined molecular surface is contributed by the CT$^{+1}$. Despite the described involvement of CT$^{+1}$ to SbC2 binding, sybodies bound to neighboring subunits do not overlap and bind remotely from the primary subunit interface and thus clearly label CALHM2 subunits engaging in homomeric interactions in the +1 position.

## Characterization of CALHM2/4 heteromers

Their high selectivity combined with the distinct location of their epitopes makes the two sybodies complementary tools for the identification of paralogs in heteromeric channels. We have thus used both binders for the structure determination of channels consisting of CALHM2 and CALHM4 subunits (CALHM2/4). To this end, we have transfected HEK293S cells with equimolar amounts of DNA encoding constructs of the respective paralogs and isolated heteromeric channels by tandem purification, sequentially capturing different tags that are attached to either construct. This strategy has allowed us to isolate channels containing both subunits, which constitute a small fraction (<10%) of the total expressed protein, most of which assemble as homomers. We have then added sybodies in stoichiometric excess and determined two datasets of CALHM2/4-sybody complexes, one containing both binders and a second SbC4 only. Both datasets are of high quality and provide insight into the organization of heteromeric channels with the sybodies defining the distribution of CALHM2 and 4 subunits (*Figure 4—figure supplements 1 and 2*, *Table 1*). In both datasets, the position of CALHM4 subunits is marked by the binding of SbC4 to the extracellular part of the protein, whereas in the dataset containing both sybodies, SbC2 additionally labels CALHM2 subunits. Both samples show a heterogeneous distribution of complexes with variable ratios of the two paralogs, which in all cases segregate into clusters to minimize heteromeric interactions (*Figure 4—figure supplements 1 and 2*). A 3D classification shows particles of three distinct oligomeric organizations. A population of proteins with pseudo D10 symmetry exhibits the characteristic organization of CALHM4 channels, which pair-wise interact on their intracellular side (*Figure 4A*, *Figure 4—figure supplement 2B*). These channels consist predominantly of CALHM4 with interspersed CALHM2 subunits (*Figure 4A*). In contrast, in two larger populations of undecameric and dodecameric channels that do not dimerize, CALHM2 subunits predominate (*Figure 4B and C*). A further classification resulted in a distribution of channels containing two to four consecutively arranged CALHM4 subunits, thus suggesting that tight homo-typic interactions between subunits prevail in heteromeric channels (*Figure 4B and C*).

## Features of heteromeric CALHM2/4 channels

Although our 3D classification has defined a broad distribution of different heteromeric assemblies of CALHM2/4 channels, the population in each distinct assembly is too small to reach high-resolution. We have thus pooled the classes of all undecameric channels containing two to four copies of CALHM4, and proceeded with further map refinement, which has allowed us to obtain maps extending to global resolutions of 3.8 Å and 3.3 Å for the datasets of samples containing either SbC4 only (CALHM2/4/SbC4), or both SbC2 and SbC4 (CALHM2/4/SbC2/SbC4), respectively (*Figure 5*, *Figure 4—figure supplements 1 and 2*, *Table 1*). The two cryo-EM densities provide congruent views of the relative arrangement and conformational preferences of CALHM2 and CALHM4 subunits in heteromeric channels. In both datasets, the two positions uniformly occupied by CALHM4 (referred to as 1 and 2 when numbered in a counterclockwise direction from the extracellular side) are well-defined and clearly discernable by SbC4 bound to its extracellular epitope (*Figure 5*). Similarly, the seven following positions (3-9), occupied by CALHM2, are well defined, whereas the two positions adjacent to the two CALHM4 subunits in a clockwise direction (11 and 10), which contain CALHM2 or 4 at different ratios are of somewhat lower resolution due to the averaging of two different subunits. This feature is particularly pronounced in the dataset of CALHM2/4/SbC2/SbC4, containing both sybodies, where position 11 bears closer resemblance to CALHM4, as further evidenced by weak residual density of SbC4 (*Figure 5D*). Conversely, features in the neighboring position 10 are closer to CALHM2, as expected for an average of channel populations containing two to four CALHM4 subunits (*Figure 5D*). In the CALHM2/4/SbC2/SbC4 complex, most of the positions occupied by CALHM2 (i.e. positions 4–8) are also distinguished by the strong density of SbC2 binding to its intracellular epitope (*Figure 5D*).

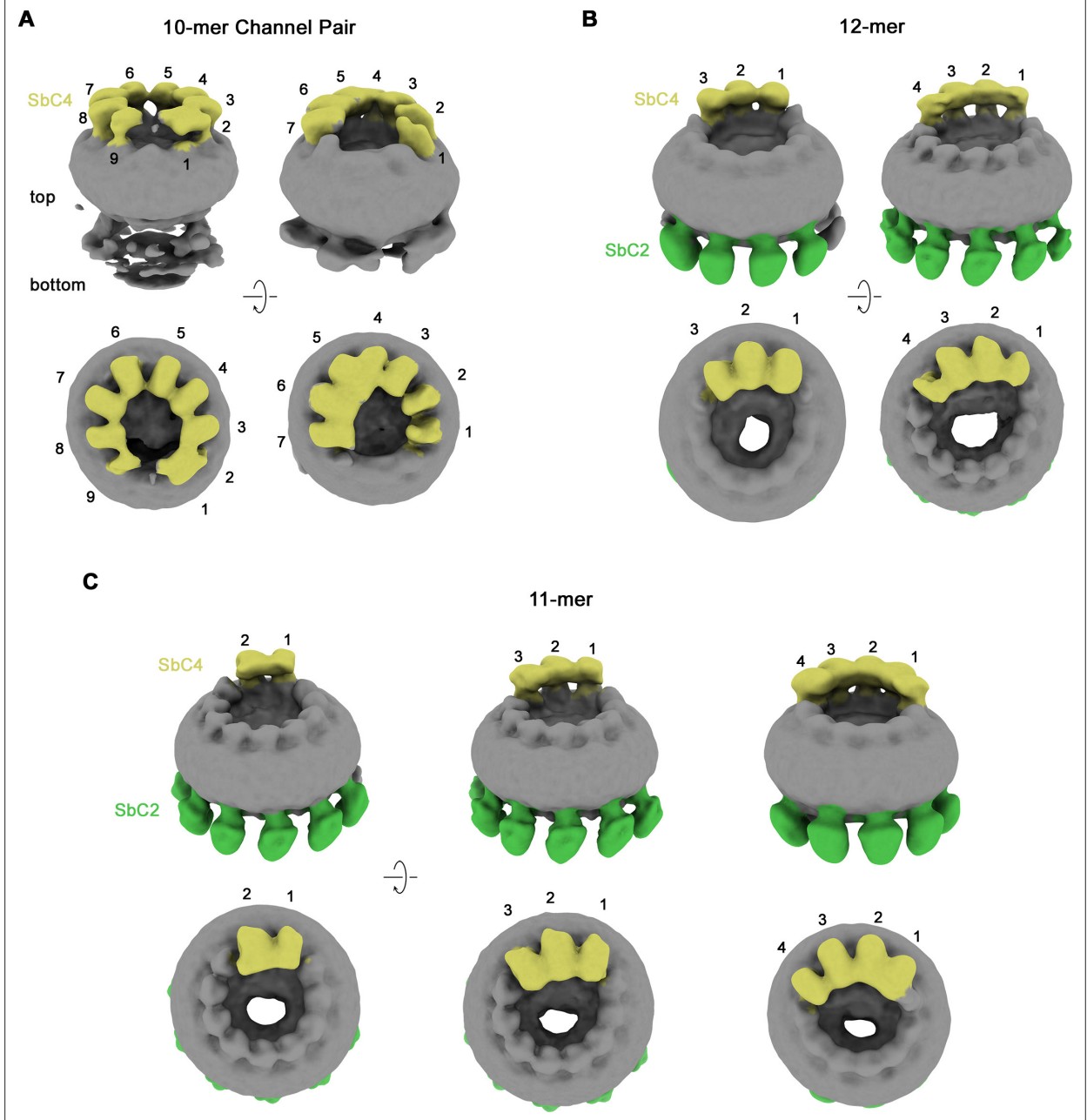

**Figure 4.** Classification of CALHM2/CALHM4 heteromers. Classes correspond to distinct populations of oligomers in the CALHM2/4/SbC2/SbC4 complex. (**A**) Assemblies containing an excess of CALHM4 subunits. Shown is the well-resolved halve of interacting channel pairs (with weak density corresponding to the second channel). (**B, C**) Assemblies with an excess of CALHM2 subunits. (**B**) Dodecameric channels containing three and four CALHM4 subunits. (**C**) Undecameric channels contain two to four CALHM4 subunits. (**A–C**) Shown are low-resolution maps (processed in C1) with bound SbC2 and SbC4 colored in green and yellow, respectively. Numbering refers to CALHM4 subunits. The relationship between views is indicated. Top panels show an inclined view towards the membrane plane, bottom panels a view from the extracellular side.

The online version of this article includes the following figure supplement(s) for figure 4:

**Figure supplement 1.** Cryo-electron microscopy (Cryo-EM) reconstruction of CALHM2/4 in complex with sybody SbC4.

**Figure supplement 2.** Cryo-electron microscopy (Cryo-EM) reconstruction of CALHM2/4 in complex with sybodies SbC2 and SbC4.

However, since the C-terminus of the neighboring CALHM2 (CT$^{+1}$) contributes to binding, we find weaker density in position 9 (*Figure 5D*). In contrast, the weaker density of SbC2 in position 3, which is fully occupied by CALHM2, is probably a consequence of the distinct conformation of this particular subunit, which compromises sybody binding.

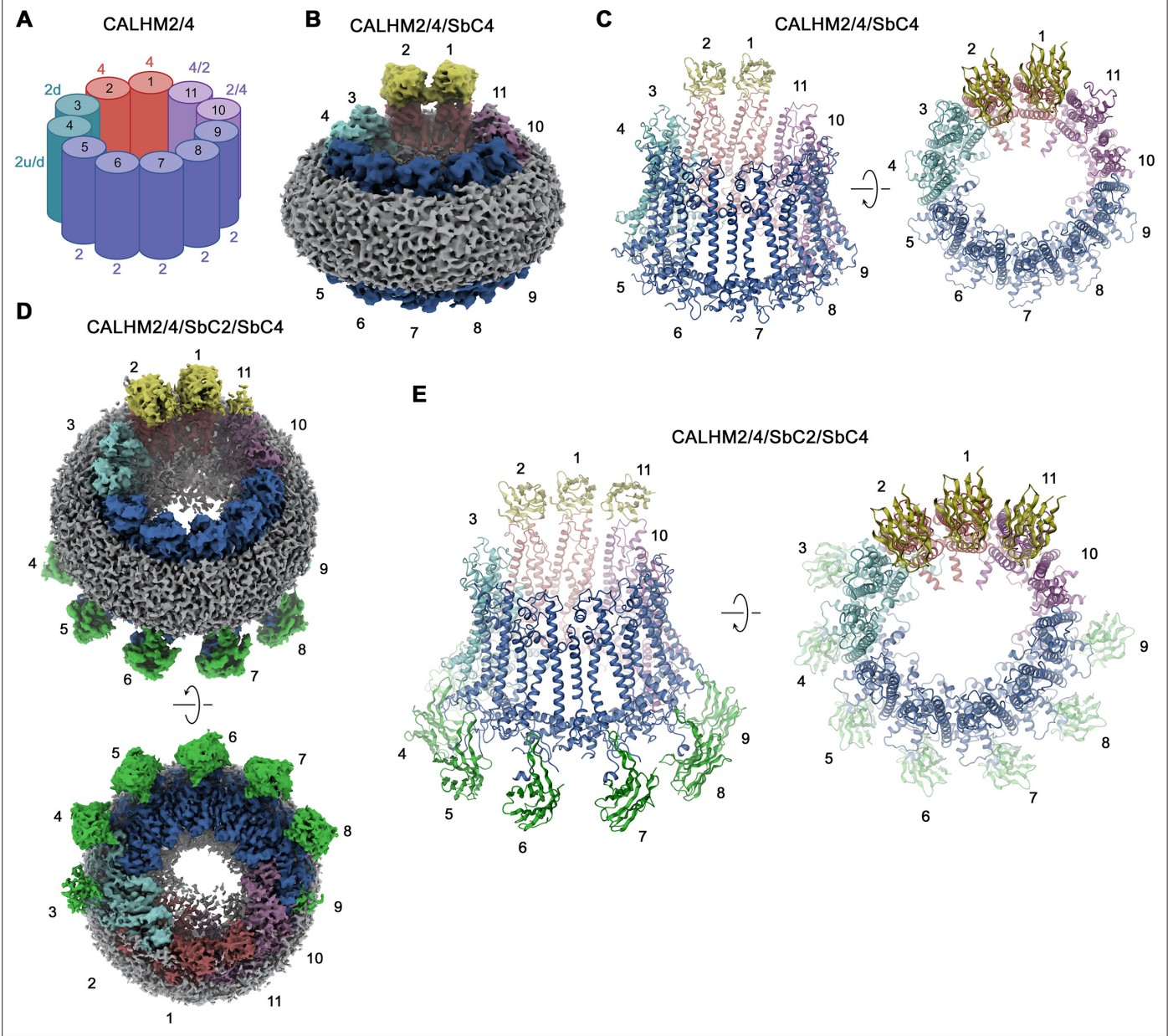

**Figure 5.** CALHM2/CALHM4 heteromer structure. (**A**) Schematic depiction of the subunit distribution in CALHM2/4 heteromers. 4 refers to CALHM4, 2 to CALHM2 in 'up' conformation, 2d to CALHM2 in 'down' conformation, 2 u/d to a mixed population of both conformations of CALHM2 subunits, 4/2, and 2/4 to mixed populations of CALHM2 and 4 with the first number indicating the predominant subunit. (**B**) Cryo-electron microscopy (Cryo-EM) density of the CALHM2/4/SbC4 complex at 3.8 Å viewed with an inclination towards the membrane plane. (**C**) Ribbon representation of the CALHM2/4/ SbC4 complex viewed from the same perspective as in B (left) and from the extracellular side (right). (**D**) Cryo-EM density of the CALHM2/4/SbC2/SbC4 complex at 3.3 Å viewed with an inclination towards the membrane plane (top) and from the intracellular side (bottom). (**C**) Ribbon representation of the CALHM2/4/SbC2/SbC4 complex viewed from the same perspective as in B (left) and from the extracellular side (right). (**B–E**), Colors, and numbers of channel subunits are as in (**A**). SbC2 and SbC4 are colored in green and yellow, respectively.

## Conformational properties of single subunits in CALHM2/4 channels

Whereas in CALHM2/4 heteromers, the CALHM4 subunits reside in the preferential 'down' conformation observed in homomeric channels, irrespective of their relative position in the channel, the conformational properties of CALHM2 subunits are influenced by their environment (*Figure 6A*). CALHM2 subunits that are distant from CALHM4 predominantly reside in the familiar 'up' position of TM1 observed in the structure of the CALHM2/SbC2 complex (*Figure 6A*). In contrast, CALHM2 in position 3 shows a distinct conformation, where TM1 has rearranged to maximize its contacts with TM3,

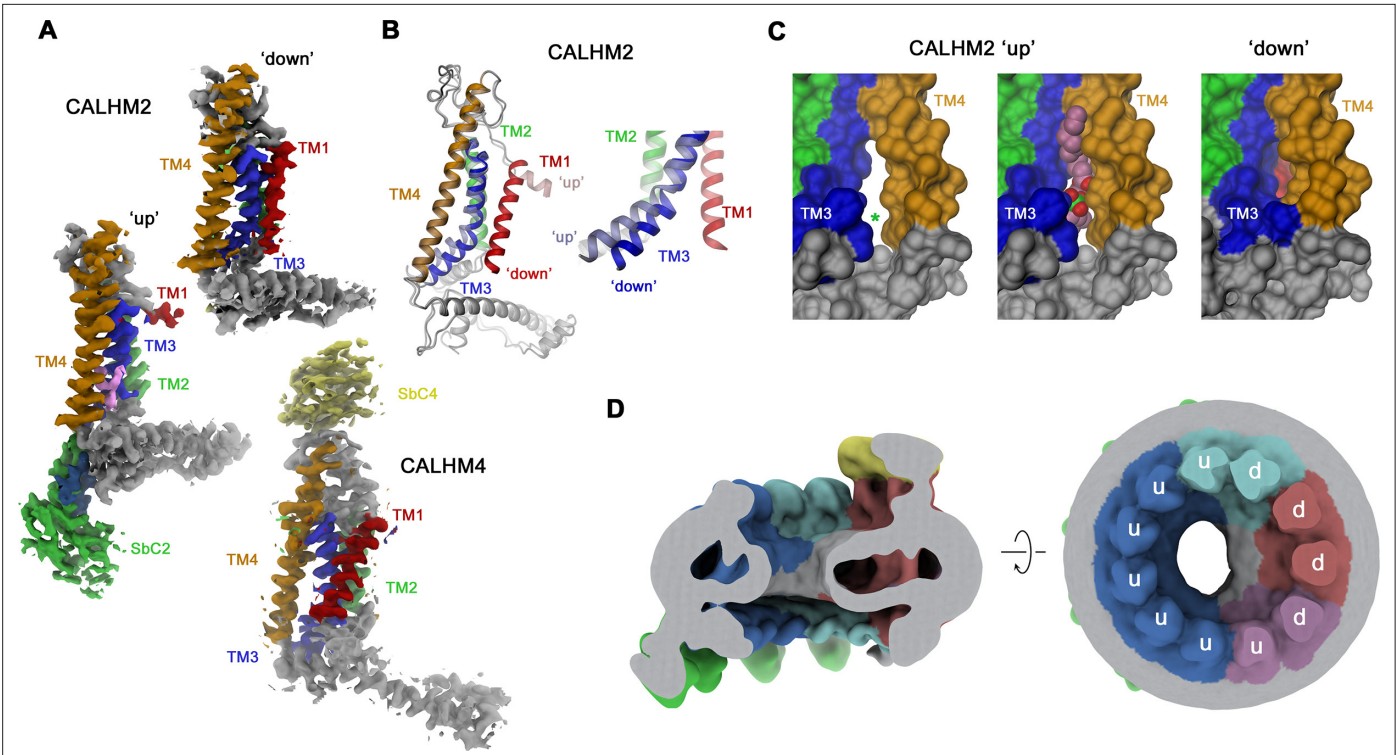

**Figure 6.** Conformational properties of calcium homeostasis modulators (CALHM) subunits in CALHM2/4 heteromers. (**A**) Cryo-electron microscopy (Cryo-EM) density of CALHM2 subunits in 'down' (position 3, top) and 'up' (position 6, left) and of CALHM4 (in 'down' conformation, position 1, bottom). (**B**) Ribbon representation of a superposition of CALHM2 subunits in 'up' and 'down' conformations. The inset (right) shows a blowup of the intracellular region around TM1-3. (**C**) Molecular surface of the intracellular region relating TM3 and TM4 viewed from within the membrane plane. A gap between both helices (left, asterisk), which is occupied by a lipid (space-filling representation) in the 'up' conformation (center), is closed in the 'down' conformation (right). (**A–C**) Transmembrane helices and attached sybodies are shown in unique colors. (**D**) Low-pass filtered map of the CALHM2/4/SbC2/SbC4 complex depicting residual bilayer-like density in the pore lumen. Slice through the channel in a view parallel to the membrane (left), and from the extracellular side (right). Subunit conformations are indicated (d, 'down', u, 'up'). CALHM4 subunits are shown in red and magenta, CALHM2 subunits in blue and cyan.

resembling the 'down' conformation of adjacent CALHM4 subunits (*Figure 6A and B*). As a consequence of this interaction, TM1 and TM3 have moved towards the outer ring of helices constituted by TM2-4 to close the fenestration between TM3 and TM4, which in the CALHM2/SbC2 complex was found to be occupied by a lipid (*Figure 3D*, *Figure 6B and C*). These movements affect the local structure of the epitope recognized by SbC2 as manifested in the weaker density observed in the CALHM2/4/SbC2/SbC4 complex, which indicates a deterioration of the interaction (*Figure 5D*). The influence of CALHM4 on CALHM2 conformations is also observed in position 4, where we find a mix of both states, and in the weak density of the 'down' conformation in CALHM2 subunits located remotely from CALHM4. The distinct preference for subunit conformations is also manifested in the distribution of residual density in the pore region, which is evident in low-pass filtered maps (*Figure 6D*). Here, we find a pronounced toroidal density surrounding subunits in a down-conformation, which contains bilayer-like features observed in CALHM4 homomers and which is absent around CALHM2 subunits sharing a predominant up-conformation (*Figures 2D, 3E and 6D*). Together, our results have defined the oligomeric organization of heteromeric channels composed of CALHM2 and CALHM4 subunits and they show how their assembly impacts the conformational preference of subunits.

## CALHM subunit interfaces

The CALHM2/4 heteromer contains four distinct subunit interfaces that provide insight into the structural basis for heteromerization. Whereas the homotypic interactions between pairs of CALHM2 or CALHM4 subunits are essentially identical as in the respective homomeric channels, the structure also reveals the well-defined interaction interface between CALHM4 and 2 subunits (viewed from the

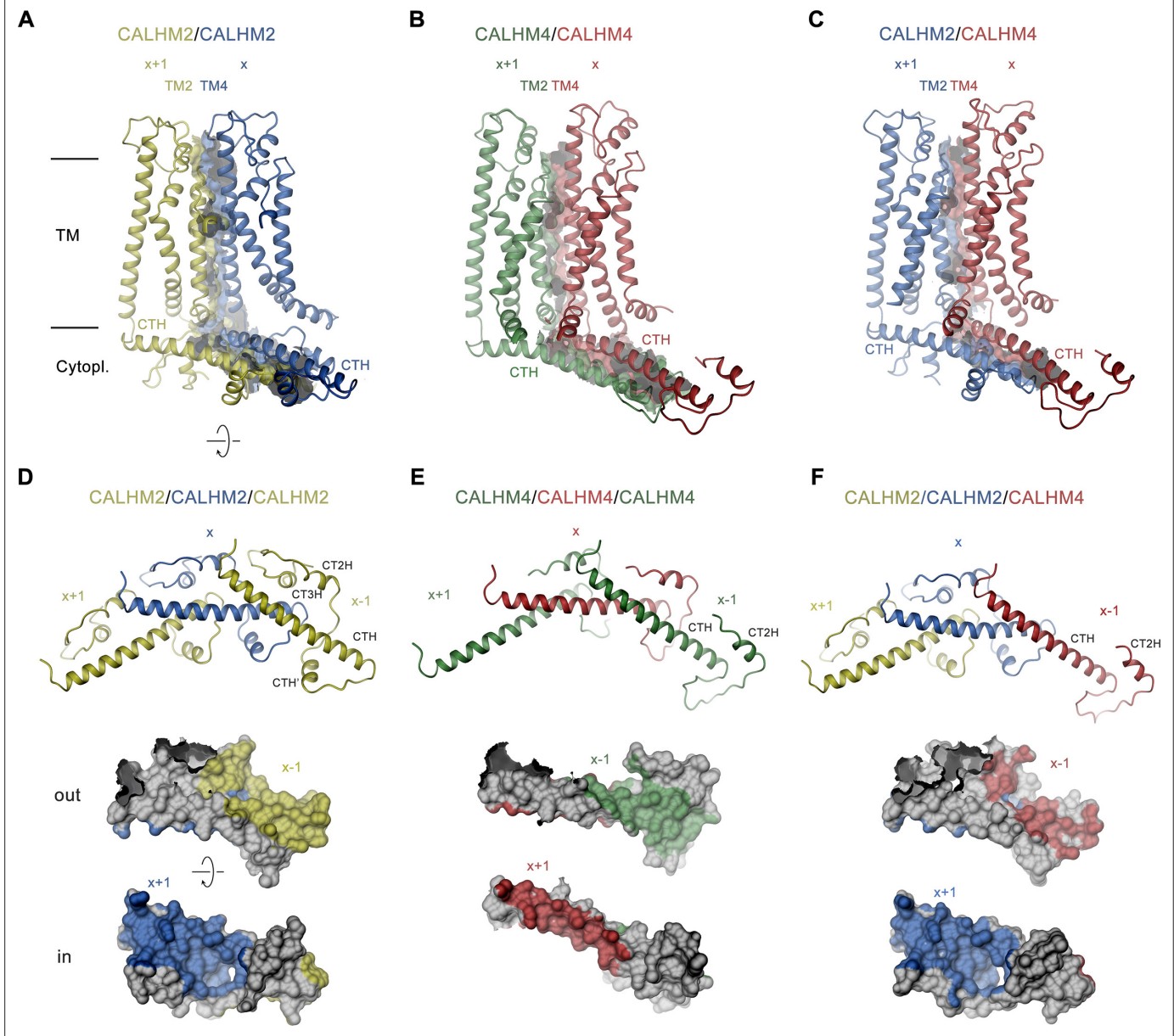

**Figure 7.** Structural features of subunit interfaces. (**A–C**) Interaction interfaces between pairs of CALHM2 subunits (**A**), CALHM4 subunits (**B**), and a CALHM2 and a CALHM4 subunit (**C**). Shown are ribbon representations of neighboring subunits viewed from the inside of the channel parallel to the membrane and their molecular surfaces that are buried in the interaction interface. (**D–F**) Subunit interactions in the cytoplasmic domain of calcium homeostasis modulators (CALHM) channels. Shown is a central subunit (**x**) and its interaction partners in clockwise and counterclockwise position (top). The region of the intracellular domain that is contacted by neighboring subunits is mapped on the molecular surface of the central subunit. Center, view from the extracellular side (out), bottom from the cytoplasm (in). (**A–F**) The relative position of subunits within the assembly is indicated. Paralogs are represented in unique colors.

extracellular side in the counterclockwise direction) whereas the details of the 2/4 interface are blurred by the mix of subunits at the 10 and 11 positions (*Figures 5 and 7*). With 5400 Å$^2$, the interface buried between two CALHM2 subunits is 40% larger compared to the corresponding CALHM4 interface (burying 3960 Å$^2$), with the latter being similar in size to the heteromeric CALHM4/CALHM2 interface (3572 Å$^2$, *Figure 7A–C*). The larger surface buried between CALHM2 subunits, which presumably translates into a stronger interaction, reflects the observed prevalence of homomers upon coexpression of subunits. In CALHM2/4 heteromers it accounts for the dominance of one paralog in different channel populations combined with the observed segregation of subunits (*Figure 4*).

The subunit interfaces consist of two separated parts, one contributed by the membrane-inserted helices TM2 and TM4, and the other by the cytoplasmic parts following TM4, comprising the helix CTH and subsequent regions (*Figure 7A–C*). Within the membrane, contacts are similar and dominated by hydrophobic residues several of which are conserved between both paralogs, although polar contacts extend further towards the extracellular region in CALHM2 (*Figure 7A*). In the cytoplasmic domain, the interactions show pronounced differences with the contact area buried by CALHM2 being considerably larger (3003 Å$^2$) compared to CALHM4 (1909 Å$^2$), which is a consequence of the parts located C-terminally to CTH, which differ in both paralogs (*Figure 7D and E*). Whereas interactions mediated by the conserved CTH are similar between CALHM2 and 4, the following region in CALHM2 consists of three short helices (CTH′, CT2H, and CT3H), and connecting loops which wrap around CTH to increase the interface and contribute to subunit interactions (*Figure 7D*). In CALHM4, of these three short helices only CT3H is present which engages in similar conserved interactions with the CTH as the equivalent region in CALHM2, although with different subunits (*Figure 7E*). While, due to the altered location of CT2H, the helix in CALHM2 primarily binds to an epitope located in the interacting subunit in the +1 position, in CALHM4 the helix binds to the equivalent region in the same subunit. A generally similar pattern of interactions is also found between subunits at the CALHM4/2 interface where the buried surface area at the C-terminus is even smaller than in CALHM4/4 interfaces (1671 Å$^2$, *Figure 7F*). In summary, we find conserved complementary contacts between CALHM2 and CALHM4 subunits, which promote the formation of heteromeric channels. However, the larger area that is buried in interfaces between equivalent subunits underlines the observed prevalence of homomeric interactions.

## Discussion

The formation of heteromers of close paralogs of a protein family is a common mechanism in eukaryotic ion channels to confer distinct functional properties to an oligomeric assembly. Heteromeric assemblies are found in pentameric ligand-gated ion channels, glutamate receptors, and volume-regulated anion channels of the LRRC8 family (*Baumann et al., 2001*; *Greger and Mayer, 2019*; *Voss et al., 2014*). In the CALHM family of large pore channels, the formation of heteromers has been demonstrated for the paralogs CALHM1 and CALHM3, where the pairing of subunits results in a strong improvement of the activation properties compared to their homomeric equivalents (*Figure 1B and C*; *Ma et al., 2018b*). We reasoned that a similar process might also extend to other family members and thus investigated the heteromerization of three paralogs that we have previously shown to be expressed in placental epithelia but for which we were so far unable to detect any function (*Drożdżyk et al., 2020*). Our experiments here have demonstrated the ability of at least two paralogs, CALHM2 and 4, to form heteromeric assemblies (*Figure 1A*). Although the existence of such mixed assemblies in a cellular context is still uncertain, our study has shed light on the detailed features of the interaction of both subunits. In that way it has revealed energetic relationships between homo- and heteromeric subunit pairing, providing insight that might also be relevant for other ion channels.

The ability to form heteromers appears specific to certain members of the CALHM family, since not all of the investigated combinations of paralogs were equally suited to interact (*Figure 1A*). The general ability of subunits to become part of the same assembly is encoded in their structure. Based on the comparison of known structures, there is an apparent incompatibility between one group consisting of CALHM1 and 3 and the second encompassing CALHM 2, 4, 5, and 6. This is reflected in the distinct orientation of the C-terminal domain, which constitutes a large part of the interaction interface (*Figure 8A and B*). In its orientation, the first helix of this domain termed CTH is similar in CALHM2, 4, 5, and 6 and, although different degrees of pairing have been detected between these subunits (*Figure 1A*), the energetic penalty to form heteromers is presumably small (*Figure 8A*). In contrast, the conformation of the C-terminus of CALHM1 and presumably also CALHM3 would likely interfere with their incorporation in a heteromeric assembly with any of the other four subunits (*Figure 8B*). In this respect, it is noteworthy that the same unit was previously shown to be responsible for the smaller oligomeric assembly of CALHM1 as either heptamer or octamer (*Ren et al., 2022*; *Syrjanen et al., 2020*). The fact that in case of the co-expression of CALHM2 and CALHM4, the homomeric populations prevail, underlines the energetic preference of homotypic interactions. Such prevalence was previously also observed upon overexpression of different members of the unrelated LRRC8 family (*Rutz et al., 2023*), where homomeric assemblies are believed not to be present in a cellular

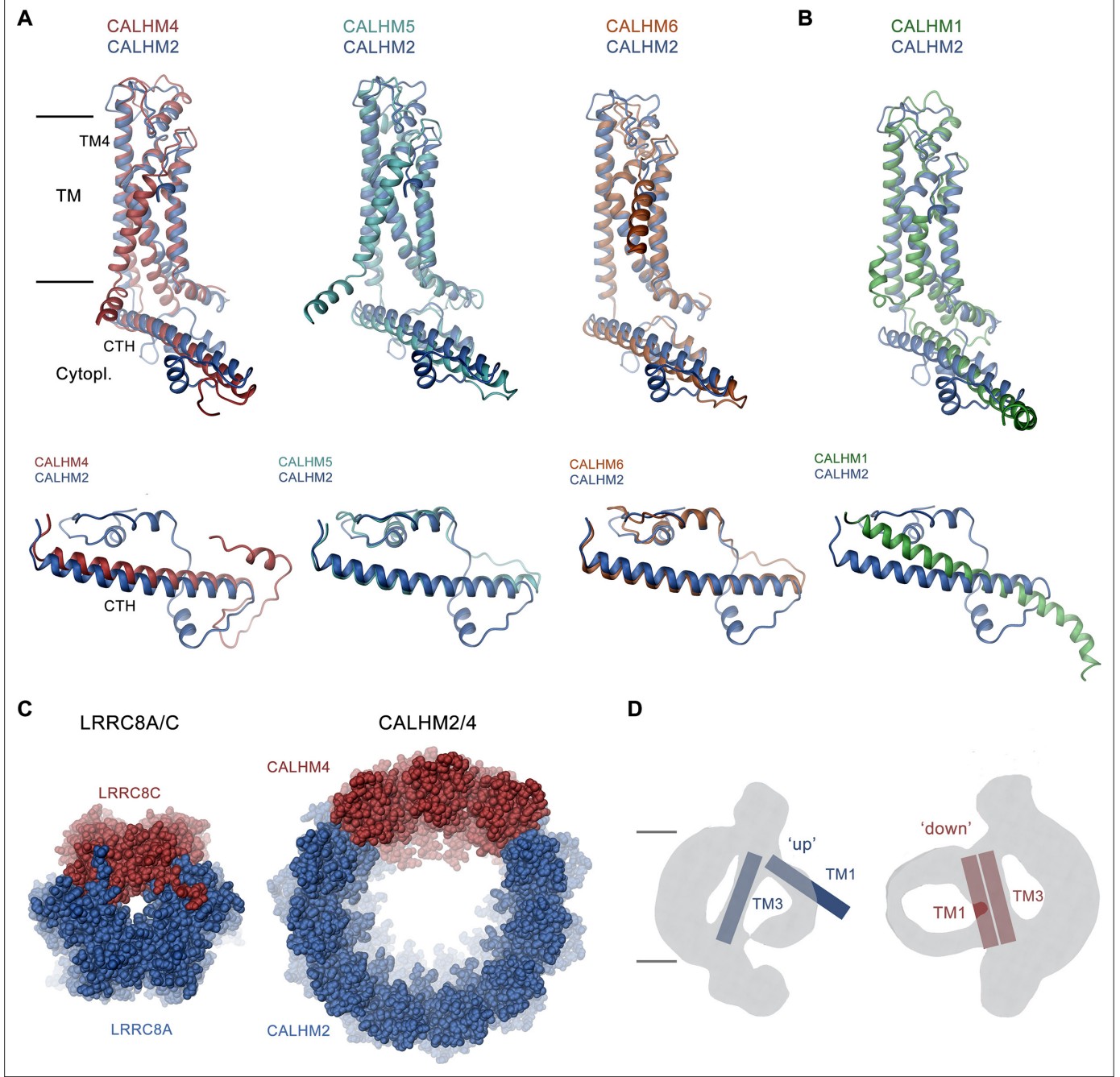

**Figure 8.** Discussion. (**A, B**) Superposition of single subunits of different calcium homeostasis modulators (CALHM) paralogs on CALHM2 viewed from within the membrane (top) and of their cytosolic domain viewed from the outside (bottom). (**A**) Superposition with CALHM2-like subunits CALHM4-6 (CALHM5 PDB:7D60, CALHM6 PDB:6YTV). (**B**) Superposition with CALHM1 (PDB:8GMQ). The difference in the conformation of the intracellular domain is apparent. (**C**) Segregation of subunits in heteromeric channels of the LRRC8 (PDB:8B41) family (left) and of CALHM2/4 channels (right). Proteins are shown as space-filling models. The view is from the extracellular side. (**D**) Schematic depiction of how the conformation of CALHM subunits influences the properties of the bilayer-like density inside the channel. The conformation of helices is indicated.

environment, pointing towards a tightly regulated oligomerization process that might be disturbed upon overexpression. However, in contrast to LRRC8A/C heteromers which, when expressed under equivalent conditions, adopt a single oligomeric organization with a 2:1 ratio of A to C subunits (*Rutz et al., 2023*), we find a broad distribution of different assemblies in the case of CALHM2/4 heteromers (*Figures 4 and 8C*). For either protein family, the predominance of one of the paralogs

and a segregation of subunits within the oligomer provides additional evidence for the higher affinity of homomeric interactions (*Figure 8C*).

Finally, we found a similar general prevalence of subunit conformations observed in homomers also in heteromeric assemblies, except that in some cases the distribution of conformations is influenced by their neighborhood. This is illustrated in the preference of CALHM2 subunits to reside in a lifted 'up' conformation of TM1 if the subunit is located distant from CALHM4 subunits whereas this conformation changes into a 'down' conformation in the vicinity of CALHM4 subunits (*Figures 5 and 6A*). The conformational preference of subunits is also extended to the shape of residual density inside the pore, which exclusively around subunits in a 'down' conformation shows the described bilayer-like features found in CALHM4 channels (*Figures 6D and 8D*). These observations underline the general independence of subunits in assuming their conformation but also indicate some coupling between neighbors. Thus, a change between conformations would not necessarily have to proceed as a fully concerted transition.

Despite the observed diversity of subunit assemblies and pore conformations in CALHM2/4 heteromers, it is puzzling not to find detectable changes in their functional properties, as neither of the constructs shows any sign of activity under conditions where CALHM1/3 heteromers robustly conduct ions (*Figure 1B–F*, *Figure 1—figure supplement 1*), despite the fact that their larger assembly should result in a wider pore diameter. The activation properties of these CALHM paralogs have thus remained elusive and might require still unidentified interaction partners. Alternatively, we also cannot exclude that these large oligomers might serve a function or substrate preference that differs from classical large pore channels, which could be modified by the observed formation of heteromers.

## Methods

### Expression constructs

CALHM coding sequences (synthesized by GeneScript) were inserted into a modified FX-cloning compatible pcDNA1.3 vector (*Drożdżyk et al., 2020*; *Geertsma, 2013*; *Geertsma and Dutzler, 2011*). The expressed CALHM constructs contained at their C-terminus either a Venus, myc and Streptavidin-Binding Peptide (Venus-myc-SBP) tag or a myc and $His_6$ (myc-His) tag. A 3C protease cleavage sequence between the CALHM and tag sequences permitted the removal of the c-terminal fusion after purification. Sybody coding sequences were cloned into the FX-compatible pSb_init vector, which contains a chloramphenicol-resistance gene and an arabinose-inducible promotor (*Zimmermann et al., 2020*). The expressed sybody constructs contained an N-terminal pelB leader sequence and a C-terminal $His_6$-tag. To obtain tag-free sybodies used in the structural characterization of heteromeric CALHM2/4 channels, sybody sequences were cloned into a pBXNPHM3 vector containing an N-terminal pelB signal sequence followed by a $His_{10}$-tag, maltose binding protein, and a 3C cleavage site.

### Cell lines

HEK293S GnTI$^-$ and HEK-293 cells were obtained from ATCC (CRL-3022 and CRL-1573) and authenticated by the provider. The cell lines tested negatively for mycoplasma contamination.

### Expression in HEK293 suspension cell culture

Human CALHM proteins were transiently expressed in suspension HEK293S GnTI$^-$ cells following previously described protocols (*Drożdżyk et al., 2020*). HEK293S GnTI$^-$ cells were cultured in HyCell HyClone TransFx-H medium (Cytiva) supplemented with 1% fetal bovine serum (FBS), 4 mM L-glutamine, 1.5% Kolliphor 188, and penicillin/streptomycin. The cells were maintained at 37 °C in a humidified incubator at a 5% $CO_2$ atmosphere at 185 rpm. One day before transfection, the cells were diluted to a density of 0.5 Mio/ml and transfected with a mixture of DNA and PEI. In co-transfections, a mixture of DNA constructs at an equivalent molar ratio was used. For 300 ml of cell culture, 0.5 mg of DNA and 1.2 mg of PEI MAX were each suspended in 16 ml of DMEM medium. DNA solutions were mixed with the PEI solutions and incubated for 15 min. To increase transfection efficiency, the mixture was supplemented with 4 mM valproic acid. The transfection mixtures were then added to cell cultures and expression was carried out for 36 hr. After this period, the cells were pelleted by centrifugation at 500 g for 25 min, washed with PBS buffer, and stored at –20 °C for further use.

## CALHM protein purification

All protein purification steps were conducted at 4 °C. HEK293S GnTI⁻ cells expressing SBP-tagged CALHM proteins for purification of homomeric channels or co-expressing SBP tagged CALHM2 and His$_6$-tagged CALHM4 for purification of heteromeric channels were harvested (typically from 1 to 2 l of cell culture for purification of homomeric channels and from 9 l for purification of heteromeric channels) and suspended at a ratio of 1:3 w/v in lysis buffer consisting of 25 mM HEPES (pH 7.6), 150 mM NaCl, 1% lauryl maltose-neopentyl glycol (LMNG), 0.5 mM CaCl$_2$, 2 mM MgCl$_2$, protease inhibitors, RNase and DNase. After 1 hr, the lysate was clarified by centrifugation at 15,000 g for 20 min. The clarified lysate was mixed with Strep-Tactin SuperFlow resin (1 ml of bed-resin per 5 g of cell pellet) and incubated for 1 hr. Following, the resin was washed with five column volumes (CV) of SEC buffer containing 10 mM HEPES (pH 7.6), 150 mM NaCl, 50 µM GDN, 2 mM CaCl$_2$. Bound proteins were eluted from the resin using 3 CV of SEC buffer supplemented with 10 mM d-desthiobiotin. For the isolation of heteromeric CALHM2/4 channels, an extra affinity chromatography step on Ni-NTA resin was performed after the initial purification on Strep-Tactin resin. To this end, the eluate from the first purification step was incubated with NiNTA resin at a 6:1 v/v ratio for 1 hr under gentle agitation. Subsequently, the resin was washed with 30 column volumes (CV) of SEC buffer, and bound proteins were eluted with 3 CV of the same buffer supplemented with 300 mM imidazole. To remove fusion tags, eluates were incubated with a 1:1 molar ratio of 3 C protease for 30 min. Samples were then concentrated to 0.5 ml using an Amicon Ultra centrifugal filter (100 kDa cutoff; Millipore), filtered through 0.22 µm filters, and subjected to size exclusion chromatography on a Superose 6 10/300 GL column equilibrated with SEC buffer. Peak fractions were pooled, concentrated using a 100 kDa MWCO centrifugal filter, and immediately used for the preparation of cryo-EM grids.

## Pulldown binding assays

For pull-downs of CALHM homologues, HEK293S GnTI⁻ cells were co-transfected using mixtures of vectors encoding one CALHM paralog containing a fusion to Venus-myc-SBP and another paralog containing a myc-His tag at equivalent molar ratios. All steps of the pulldown binding assays were performed at 4 °C. Cells from a 300 ml culture were lysed in lysis buffer (25 mM HEPES (pH 7.6), 150 mM NaCl, 1% LMNG, protease inhibitors, RNase, DNAse, 0.5 mM CaCl$_2$, 2 mM MgCl$_2$) for 1 hr and the lysates were cleared by centrifugation at 15,000 rcf for 10 min. The supernatants were filtered through a 5 µm filter, mixed with 2 ml of Strep-Tactin SuperFlow resin, and incubated for 2 hr under gentle agitation. The resin was then washed with 5 CV of SEC buffer and bound proteins were eluted with elution buffer (SEC buffer supplemented with 10 mM d-desthiobiotin, pH 7.6). For Western blot analysis, samples were loaded on a 4–20% SDS-polyacrylamide gel and transferred to a polyvinylidene fluoride membrane by a semi-dry blotting procedure. The membranes were first blocked at room temperature for 1 hr with 5% non-fat milk in TBST buffer (50 mM Tris (pH 7.6), 150 mM NaCl, 0.075% Tween20) and then incubated with anti-Myc primary antibodies overnight at 4 °C. The membranes were then washed three times with TBST and incubated with horse-radish peroxidase-conjugated goat anti-mouse IgG antibody for 2 hr at 4 °C. After blotting, the membranes were washed three times with TBST buffer and developed with the Amersham ECL Prime Western Blotting Detection kit (GE Healthcare).

## Sybody selection

For sybody selection procedures, CALHM2 and CALHM4 proteins were chemically biotinylated using amine-reactive EZ-Link NHS-PEG4-Biotin (Thermo Fisher Scientific). For this purpose, proteins were diluted to a concentration of 25 µM in SEC buffer, and the coupling agent was added at an eight-fold molar excess. Samples were incubated for 1 hr on ice and the reaction was quenched by the addition of 5 mM Tris-HCl (pH 8.0). Excess biotin was removed from the biotinylated protein on a PD-10 desalting column. The eluted proteins were concentrated, mixed with glycerol to a final concentration of 10%, aliquoted, flash-frozen in liquid N$_2$, and stored at –80 °C. Sybody selection was performed as previously described (*Zimmermann et al., 2020*). Three synthetic ribosome-display libraries, concave, loop, and convex libraries, and the plasmids, were kindly provided by Prof. Markus Seeger. Sybodies were selected against chemically biotinylated CALHM proteins. In brief, one round

of ribosome display was performed, and the output RNA was reverse transcribed and cloned into a phage-display compatible vector. The obtained plasmids were transformed into *E. coli* SS320 cells to produce phages using a helper phage M13KO7. The produced phages were then used in the subsequent two rounds of phage display. Phagemids were isolated from the output phages of the second round, and the sequences coding for sybodies were subcloned into an pSB_init expression vector. The resulting plasmids were transformed into *E. coli* MC1061, which were plated on agar plates. Single clones were expressed on a small scale and used for binding tests by ELISA. Positive clones were sequenced.

## Sybody expression and purification

DNA constructs of sybodies were transformed into *E. coli* MC1061 by a heat shock procedure and grown in one liter of Terrific Broth medium supplemented with 25 µg/ml chloramphenicol at 37 °C. After initial incubation at 37 °C for 2 hr, the temperature was lowered to 22 °C and once the cultures reached an $OD_{600}$ of 0.5, L-arabinose was added to a final concentration of 0.02% to induce expression. After 16–18 hr of expression, bacteria were harvested by centrifugation at 8000 g for 20 min and pellets were either used immediately or stored at –80 °C until further use. For purification, pellets were resuspended in TBS buffer (50 mM Tris (pH 7.6), 150 mM NaCl) supplemented with 0.1 mg/ml lysozyme at a 1:30 w/v ratio. Bacteria were lysed by three freeze-thaw cycles followed by sonication to release the periplasmic content. Lysates were centrifuged at 8000 g for 30 min at 4 °C. The supernatants were collected and mixed with Ni-NTA resin and incubated for 30 min. The resin was washed with TBS and sybodies were eluted with TBS supplemented with 300 mM imidazole. For removal of the affinity tag, eluates were incubated overnight with 3 C protease while dialyzing with a MWCO of 5 kDa. The eluates were then concentrated and filtered through a 0.22 µm centrifugal filter and subjected to size exclusion chromatography on a Sepax SRT-10C SEC100 column equilibrated in TBS. Fractions of the monomeric sybody peaks were collected and stored at –80 °C for further use.

## Binding assays

Purified CALHM proteins were mixed with sybodies in a molar ratio of 1:2 to a final concentration of 25 µM CALHM proteins. Prior to mixing, sybodies were supplemented with GDN to a final concentration of 50 µM. Protein mixtures were applied to a Superose 6 5/150 Increase column equilibrated with SEC buffer. Peak fractions corresponding to CALHM proteins were collected, concentrated using Amicon Ultra 0.5 ml centrifugal filters (10 kDa cut-off, Ultracel), and analyzed by SDS-PAGE. SPR measurements were performed at the Functional Genomic Centre Zurich using a Biacore T200 instrument (GE Healthcare). The chemically biotinylated CALHM proteins were immobilized on a streptavidin-coated sensor chip (XanTec), resulting in a maximum response value of approximately 600 response units (RU). Measurements were performed at 10 °C at a flow rate of 30 µl/min. The buffer used during the experiments consisted of 10 mM HEPES at pH 7.6, 150 mM NaCl, 2 mM $CaCl_2$, and 0.006% GDN for CALHM2 or 0.01% LMNG for CALHM4. Flow cell 1 was left empty to act as a reference cell for the measurements. Prior to measurements, the system was equilibrated with the appropriate buffer for 2 hr. The sybodies were injected in a single cycle kinetic measurement at concentrations of 7.8, 15.6, 31.25, 125, and 500 nM for sybody SbC2 and 8, 24, 72, 216, and 648 nM for sybody SbC4. Binding data was analyzed using BIAevaluation software (GE Healthcare) and fitted to a single-site binding model.

## Whole-cell patch-clamp recordings

Adherent HEK-293 cells were gently detached from their support and seeded in 10 cm dishes at 10% confluency. After a 3 hr incubation, cells were transfected with a mixture of 12 µg PEI and 3 µg of plasmids encoding CALHM constructs, which were C-terminally tagged with Venus-Myc-SBP. For co-expression experiments, cells were transfected at a 1:1 ratio of the constructs at a total amount of 3 µg of plasmids. Mock cells were transfected with a vector expressing Venus tag only. Prior to transfection, DNA and PEI were separately incubated in 0.5 ml of DMEM medium. After 5 min, both solutions were mixed, incubated for 15 min, and added to respective cell cultures. Whole-cell currents were recorded 16–24 hr after transfection at room temperature (20–22°C). Recordings were conducted using borosilicate glass capillaries that were pulled and polished to achieve a final resistance of 4–6 MΩ. Patches were held at –40 mV and the voltage was altered between –80 mV and +100 mV in 10 mV increments

for 500 ms by a step protocol. Cells were perfused using a gravity-fed system. The solutions were as previously described (*Syrjanen et al., 2020*). The pipette solution contained 147 mM NaCl, 10 mM EGTA, and 10 mM HEPES at pH 7.0, using NaOH to adjusting the pH. The bath solution contained 147 mM NaCl, 13 mM glucose, 10 mM HEPES (pH 7.3), using NaOH for adjusting the pH, 2 mM KCl, 2 mM CaCl$_2$, and 1 mM MgCl$_2$. In Ca$^{2+}$-free experiments, the bath solution was similar but without the addition of CaCl$_2$. Data acquisition was performed using an Axopatch 200B amplifier and either Digidata 1440 or 1550 (Molecular Devices). Analog signals were digitized at a sampling rate of 10–20 kHz and filtered at 5 kHz using the built-in four-pole Bessel filter. Clampex 10.6 software (Molecular Devices) was used for data acquisition and GraphPad Prism 8 for data analysis.

## Cryo-EM sample preparation and data collection

Cryo-EM grids were prepared with freshly purified CALHM proteins. Prior to freezing, sybodies were added to the CALHM samples at a 1:1.5 molar excess (based on the total amount of CALHM subunits). In the sample containing heteromeric CALHM2/4 channels and both sybodies, SbC4 was added at a 1.5 and SbC2 at a 2.5 molar excess. The samples were frozen at different concentrations of CALHM proteins (20 µM in the CALHM2/SbC2 sample, 47 µM in the CALHM4/SbC4 sample, 38 µM in the CALHM2/4/SbC4 sample, and 45 µM in the CALHM2/4/SbC2/SbC4 sample). Samples were applied to glow-discharged holey carbon grids (Quantifoil R1.2/1.3 Au 200 mesh). Excess liquid was removed in a controlled environment (4 °C and 100% relative humidity) by blotting grids for 2–4 s. Grids were subsequently flash-frozen in a liquid propane-ethane mix using a Vitrobot Mark IV (Thermo Fisher Scientific) and stored in liquid nitrogen. Samples were imaged with a 300 kV Titan Krios G3i with a 100 µm objective aperture and data were collected using a post-column BioQuantum energy filter with a 20 eV slit and a K3 direct electron detector operating in super-resolution counted mode. Dose-fractionated micrographs were recorded with a defocus range of –1 µm to –2.4 µm in an automated mode using EPU 2.9. Datasets were recorded at a nominal magnification of 130,000 corresponding to a pixel size of 0.651 Å (0.3255 Å per pixel in super-resolution mode).

## Cryo-EM image processing

Micrographs from all datasets were pre-processed in a similar manner. Micrographs of the CALHM2/SbC2, CALHM2/4/SbC4, and CALHM2/4/SbC2/SbC4 datasets were binned prior to processing resulting in a pixel size of 0.651 Å. The raw movies were used to correct for beam-induced motion using patch motion correction in cryoSPARC v.3.2.0 and v.4.0.3, followed by patch CTF estimation (*Punjani et al., 2017*). For the CALHM4/SbC4 dataset, the raw movies were used for correction of the beam-induced movement using a dose-weighting scheme in RELION's own implementation of the MotionCor2 algorithm available in version 3.0 (*Zivanov et al., 2018*) and the CTF parameters were estimated on summed movie frames using CTFFIND4.1 (*Rohou and Grigorieff, 2015*). Poor quality images with significant drift, ice contamination or poor CTF estimates were discarded.

The dataset of CALHM4/SbC4 was processed entirely in RELION-3.1 (*Scheres, 2012*). Particles were first selected by Laplacian-of-Gaussian-based auto-picking and subjected to a first round of 2D classification to generate 2D average templates. Next, four 2D classes with protein-like features were used as references for a second template-based picking. The particles were extracted with a box size of 720 pixels and downscaled four times. A second 2D classification was performed and the particles from the five best classes were used for the generation of an initial 3D model, imposing C10 symmetry. The extracted particles from the second 2D classification round were subjected to a round of 3D classification with C10 symmetry imposed, using the initial model low pass filtered to 60 Å as a reference map. The best 3D class was selected and the particles belonging to this class were unbinned to their original pixel size and subjected to 3D refinement with D10 symmetry imposed. Additionally, CTF and aberration refinement, and Bayesian polishing were applied (*Rohou and Grigorieff, 2015*; *Zivanov et al., 2019*). The map was sharpened with an isotropic b-factor of –153 Å$^2$, resulting in a global resolution of 3.7 Å. The local resolution was estimated using RELION (*Zivanov et al., 2018*).

The dataset of CALHM2/SbC2 was processed entirely in cryoSPARC v.3.2.0 and v.4.0.3 (*Punjani et al., 2017*). Particles were initially picked with the blob picker and subjected to 2D classification to generate 2D average templates for more accurate template-based particle picking. Particles were selected and extracted with a box size of 512 pixels and downscaled twice. Selected particles were subjected to four rounds of 2D classification to remove false positives and low-quality particles and

subjected to an *ab initio* reconstruction with three classes. The obtained classes included decameric, undecameric, and ambiguous assemblies of CALHM2. Particles belonging to classes showing an undecameric assembly were further sorted by a round of heterogeneous refinement using three *ab initio* reconstructions as references. The same procedure was used for particles belonging to classes showing a decameric assembly, which yielded a low-quality reconstruction. Therefore, further processing was restricted to the more promising undecameric assemblies. The quality of the reconstruction was further improved by CTF global and local refinements and non-uniform refinement with C11 symmetry imposed, yielding a map at 3.07 Å. The local resolution was estimated with cryoSPARC. To improve the resolution of the CALHM-sybody interface region, a local refinement with C11 symmetry imposed was performed in the presence of a tight mask restricted to the region surrounding the sybody binding site, yielding a reconstruction of this region at a global resolution of 2.8 Å.

The datasets of CALHM2/4 in complex with sybodies were processed in cryoSPARC v.4.0.3 (*Punjani et al., 2017*), unless indicated otherwise. Particles from the CALHM2/4 datasets containing either both sybodies (CALHM2/4/SbC2/SbC4) or SbC4 alone (CALHM2/4/SbC4) were initially picked with the blob picker and subjected to 2D classification to generate 2D templates for more accurate template-based particle picking. Particles were selected and extracted with a box size of 512 pixels, downscaled twice, and subjected to multiple rounds of 2D classification to remove false positives and poor-quality particles. Visual inspection of the 2D classes indicated the presence of different assemblies. To separate these assemblies, particles were exported to RELION (version 4.0) (*Kimanius et al., 2021*) and used to generate initial models. Subsequently, particles were subjected to several rounds of 3D classification using the initial models as references. In the first two rounds of classification, low-quality particles were excluded and only particles belonging to the high-quality classes were subjected to a final 3D classification with 10 distinct classes. To obtain high-resolution reconstructions of complexes, particles belonging to 2D classes of channel dimers were excluded and only particles belonging to high-quality 2D classes with single channels were used to generate *ab initio* reconstructions in cryoSPARC. This yielded decoy reconstructions and undecameric reconstructions. Particles of undecameric reconstructions were used as input for heterogeneous refinements with three references including a decoy class and two proper classes. This approach yielded reconstructions of undecameric and dodecameric assemblies. The heterogeneous refinement was repeated four times using particles belonging to the undecameric assembly from the previous refinement as input and undecameric and dodecameric reconstructions as references. Resulting undecameric reconstructions were further improved by local and global CTF, and non-uniform refinements in C1. These approaches yielded maps with global resolutions of 3.8 Å and 3.3 Å for the CALHM2/4/SbC4 and CALHM2/4/SbC2/SbC4 complexes, respectively. The local resolution of maps was estimated within cryoSPARC.

## Model building and Refinement

Models were built using Coot (*Emsley and Cowtan, 2004*) based on previously determined structures of CALHM2 (PDB:6UIW) (*Choi et al., 2019*) and CALHM4 (PDB:6YTL) (*Drożdżyk et al., 2020*). Sybodies were initially modeled based on high-resolution structures (PDB:1ZVH for SbC2 and PDB:3K1K for SbC4). Models were first fitted into their corresponding densities using ChimeraX (*Pettersen et al., 2021*). The cryo-EM density of CALHM2/SbC2 allowed us to assign residues 39–308 and 314–323 of CALHM2 and 324–329 of the remaining sequence of the expression tag, and the entire model of SbC2 whose CDRs interacting with CALHM2 were rebuilt into the cryo-EM density. The cryo-EM density of CALHM4/SbC4 was of sufficiently high-resolution to unambiguously assign residues 5–82 and 94–276 of CALHM4. In contrast, the limited quality of the density of SbC4 did not permit a complete interpretation of its atomic model and thus a truncated version lacking the variable regions was placed into the density.

The model of the heteromeric CALHM2/4 in complex with sybodies SbC2 and SbC4 was built by placing eight CALHM2 and three CALHM4 subunits into the cryo-EM density obtained from the CALHM2/4/SbC2/SbC4 dataset at 3.3 Å. Coordinates of CALHM subunits and sybodies were obtained from the structures of the homomeric CALHM2/SbC2 and CALHM4/SbC4 complexes. Residues 19–46 of CALHM2 in the downward conformation were built de novo. The subunit displaying a mix of CALHM2 and 4 (i.e. position 10) was interpreted by a CALHM2 subunit where loop regions not defined in the density were removed. The structure of the CALHM2/4/SbC4 complex at 3.8 Å was built based on the model of CALHM2/4/SbC2/SbC4 after the removal of the SbC2 models, except

that it contains a truncated version of CALHM2 also in position 11. Refinement and validation of all models was performed in PHENIX (*Afonine et al., 2018*). Figures containing molecular structures and densities were prepared with DINO (http://www.dino3d.org), PyMOL (*DeLano, 2002*), Chimera (*Pettersen et al., 2004*) and ChimeraX (*Pettersen et al., 2021*). Interaction interfaces were calculated with CNS_SOLVE (*Brunger, 2007*).

## Acknowledgements

This research was supported by a grant from the Swiss National Science Foundation (No. 310030B_182828). Cryo-EM data was collected at the Center for Microscopy and Image Analysis (ZMB) of the University of Zurich. We thank Markus Seeger for providing access to the sybody libraries. Jens Sobek and Marton Liziczai are acknowledged for help with surface plasmon resonance experiments recorded at the Functional Genomics Center of UZH/ETH Zurich. All members of the Dutzler lab are acknowledged for their help at various stages of the project.

## Additional information

### Funding

| Funder | Grant reference number | Author |
| --- | --- | --- |
| Schweizerischer Nationalfonds zur Förderung der Wissenschaftlichen Forschung | 310030B_182828 | Raimund Dutzler |

The funders had no role in study design, data collection and interpretation, or the decision to submit the work for publication.

### Author contributions

Katarzyna Drożdżyk, Conceptualization, Data curation, Formal analysis, Investigation, Writing - original draft, Writing - review and editing; Martina Peter, Investigation, Visualization, Writing - review and editing; Raimund Dutzler, Conceptualization, Formal analysis, Supervision, Funding acquisition, Validation, Writing - original draft, Project administration, Writing - review and editing

### Author ORCIDs

Katarzyna Drożdżyk  http://orcid.org/0000-0001-6288-4735
Raimund Dutzler  http://orcid.org/0000-0002-2193-6129

Reviewer #1 (Public review): https://doi.org/10.7554/eLife.96138.3.sa1
Reviewer #2 (Public review): https://doi.org/10.7554/eLife.96138.3.sa2
Author response https://doi.org/10.7554/eLife.96138.3.sa3

## Additional files

### Supplementary files

• MDAR checklist

### Data availability

The cryo-EM density maps have been deposited in the Electron Microscopy Data Bank under following ID codes: EMD-19365 (CALHM4/SbC4), EMD-19362 (CALHM2/SbC2), EMD-19363 (CALHM2/CALHM4/SbC4), EMD-19364 (CALHM2/CALHM4/SbC2/SbC4). The coordinates of the corresponding atomic models have been deposited in the Protein Data Bank under ID codes 8RMN (CALHM4/SbC4), 8RMK (CALHM2/SbC2), 8RML (CALHM2/CALHM4/SbC4), 8RMM (CALHM2/CALHM4/SbC2/SbC4).

The following datasets were generated:

| Author(s) | Year | Dataset title | Dataset URL | Database and Identifier |
|---|---|---|---|---|
| Peter M, Drozdzyk K, Dutzler R | 2024 | Cryo-EM structure of a dimer of decameric human CALHM4 in complex with synthetic nanobody SbC4 | https://www.rcsb.org/structure/8RMN | RCSB Protein Data Bank, 8RMN |
| Peter M, Drozdzyk K, Dutzler R | 2024 | Cryo-EM structure of a dimer of decameric human CALHM4 in complex with synthetic nanobody SbC4 | https://www.ebi.ac.uk/emdb/EMD-19365 | Electron Microscopy Data Bank, EMD-19365 |
| Drozdzyk K, Dutzler R | 2024 | Cryo-EM structure of human CALHM2 in complex with synthetic nanobody SbC2 | https://www.rcsb.org/structure/8RMK | RCSB Protein Data Bank, 8RMK |
| Drozdzyk K, Dutzler R | 2024 | Cryo-EM structure of human CALHM2 in complex with synthetic nanobody SbC2 | https://www.ebi.ac.uk/emdb/EMD-19362 | Electron Microscopy Data Bank, EMD-19362 |
| Drozdżyk K, Dutzler R | 2024 | Structure of heteromeric CALHM2/4 channel in complex with synthetic nanobody SbC4 | https://www.rcsb.org/structure/8RML | RCSB Protein Data Bank, 8RML |
| Drozdżyk K, Dutzler R | 2024 | Structure of heteromeric CALHM2/4 channel in complex with synthetic nanobody SbC4 | https://www.ebi.ac.uk/emdb/EMD-19363 | Electron Microscopy Data, EMD-19363 |
| Drozdzyk K, Dutzler R | 2024 | Structure of heteromeric CALHM2/4 channel in complex with synthetic nanobodies SbC2 and SbC4 | https://www.rcsb.org/structure/8RMM | RCSB Protein Data Bank, 8RMM |
| Drozdzyk K, Dutzler R | 2024 | Structure of heteromeric CALHM2/4 channel in complex with synthetic nanobodies SbC2 and SbC4 | https://www.ebi.ac.uk/emdb/EMD-19364 | Electron Microscopy Data Bank, EMD-19364 |

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

# Appendix 1

**Appendix 1—key resources table**

| Reagent type (species) or resource | Designation | Source or reference | Identifiers | Additional information |
|---|---|---|---|---|
| Strain, strain background (*Escherichia coli*) | MC1016 | Sigma | C66303 | |
| Cell line (*Homo-sapiens*) | HEK293S GnTI⁻ | ATCC | CRL-3022 | |
| Cell line (*Homo-sapiens*) | HEK-293 | ATCC | CRL-1573 | |
| Antibody | Anti-c-Myc (Mouse monoclonal) | Sigma | M4439 | (1:5000) |
| Antibody | Peroxidase AffiniPure Goat Anti-Mouse IgG H+L (Goat polyclonal) | Jackson ImmunoResearch | 115-035-146 | (1:10000) |
| Recombinant DNA reagent | Modified mammalian pcDNA 3.1 $^{(+)}$ expression vector for FX cloning system, C-terminal 3 C cleavage site, myc tag and 6x HisTag | Dutzler group | N/A | |
| Recombinant DNA reagent | Modified mammalian pcDNA 3.1 $^{(+)}$ expression vector for FX cloning system, C-terminal 3 C cleavage site, Venus fluorescent tag, myc tag and SBP tag | Dutzler group | N/A | |
| Recombinant DNA reagent | pBXNPHM3 (plasmid) | Seeger group | Addgene #110099 | |
| Recombinant DNA reagent | pSbinit (plasmid) | Seeger group | Addgene #110100 | |
| Recombinant DNA reagent | *Homo sapiens* CALHM1 | GenScript | Accession NM_001001412.3 | |
| Recombinant DNA reagent | *Homo sapiens* CALHM2 | GenScript | Accession NM_015916.5 | |
| Recombinant DNA reagent | *Homo sapiens* CALHM3 | GenScript | Accession NM_001129742.2 | |
| Recombinant DNA reagent | *Homo sapiens* CALHM4 | GenScript | Accession NM_001366078.1 | |
| Recombinant DNA reagent | *Homo sapiens* CALHM6 | GenScript | Accession NM_001010919.3 | |
| Commercial assay or kit | Amersham ECL Prime Western Blotting Detection Kit | GE Healthcare | RPN2232 | |
| Commercial assay or kit | EZ-link NHS-PEG4-biotin | Sigma | A39259 | |
| Chemical compound, drug | Benzamidine | Sigma | B6506 | |
| Chemical compound, drug | Calcium chloride | Sigma | 223506 | |
| Chemical compound, drug | Chloramphenicol | Sigma | C1919 | |
| Chemical compound, drug | D-desthiobiotin | Sigma | D1411 | |
| Chemical compound, drug | DNase I | Sigma | EN0521 | |

*Appendix 1 Continued on next page*

*Appendix 1 Continued*

| Reagent type (species) or resource | Designation | Source or reference | Identifiers | Additional information |
|---|---|---|---|---|
| Chemical compound, drug | Dulbecco's Modified Eagle's Medium (DMEM) High glucose, pyruvate | Sigma | D6429 | |
| Chemical compound, drug | EGTA | Sigma | 03777 | |
| Chemical compound, drug | Fetal bovine serum | Sigma | F7524 | |
| Chemical compound, drug | Glucose | AppliChem | A1422.1000 | |
| Chemical compound, drug | Glycerol 99% | Sigma | G7757 | |
| Chemical compound, drug | Glycol-diosgenin (GDN) | Anatrace | GDN101 | |
| Chemical compound, drug | HCl | Millipore | 1.00319.1000 | |
| Chemical compound, drug | HEPES | Sigma | H3375 | |
| Chemical compound, drug | HyClone HyCell TransFx-H medium | Cytiva | SH30939.02 | |
| Chemical compound, drug | Imidazole | Roth | X998.4 | |
| Chemical compound, drug | Kolliphor P188 | Sigma | K4894 | |
| Chemical compound, drug | L-(+)-arabinose | Sigma | A3256 | |
| Chemical compound, drug | L-glutamine | Sigma | G7513 | |
| Chemical compound, drug | Lauryl Maltose Neopentyl Glycol (LMNG) | Anatrace | NG310 | |
| Chemical compound, drug | Leupeptin | AppliChem | A2183.0100 | |
| Chemical compound, drug | Lysozyme | AppliChem | A3711.0050 | |
| Chemical compound, drug | Magnesium chloride | Fluka | 63.065 | |
| Chemical compound, drug | Penicillin-streptomycin | Sigma | P0781 | |
| Chemical compound, drug | Pepstatin A | Axon lab | A2205.0100 | |
| Chemical compound, drug | Phenylmethylsulfonyl fluoride (PMSF) | Sigma | PMSF-RO | |
| Chemical compound, drug | Polyethylenimine HCl MAX, Linear, MW 40,000 (PEI MAX 40000) | Chemie Brunschwig AG | POL24765 | |
| Chemical compound, drug | Potassium chloride | Sigma | 746346 | |
| Chemical compound, drug | RNase | Sigma | R5125 | |
| Chemical compound, drug | Sodium chloride | Sigma | 71380 | |

*Appendix 1 Continued on next page*

*Appendix 1 Continued*

| Reagent type (species) or resource | Designation | Source or reference | Identifiers | Additional information |
|---|---|---|---|---|
| Chemical compound, drug | Sodium hydroxide | Sigma | S8045 | |
| Chemical compound, drug | Terrific broth | Sigma | T9179 | |
| Chemical compound, drug | Tris | AppliChem | A1379 | |
| Chemical compound, drug | Tween 20 | Sigma | 93773 | |
| Chemical compound, drug | Valproic acid sodium salt | Sigma | P4543 | |
| Software, algorithm | Axon Clampex 10.6 | Molecular Devices | N/A | |
| Software, algorithm | Axon Clampfit 11.0.3 | Molecular Devices | N/A | |
| Software, algorithm | Chimera 1.16 | *Pettersen et al., 2004* | http://www.cgl.ucsf.edu/chimera/ | |
| Software, algorithm | ChimeraX 1.3 | *Pettersen et al., 2021* | https://www.cgl.ucsf.edu/chimerax/ | |
| Software, algorithm | Coot 0.9.8.91 | *Emsley and Cowtan, 2004* | https://www2.mrc-lmb.cam.ac.uk/personal/pemsley/coot/ | |
| Software, algorithm | cryoSPARC v3.2.0–4.0 | Structural Biotechnology Inc. | RRID:SCR_016501 | |
| Software, algorithm | CTFFIND4.1 | *Rohou and Grigorieff, 2015* | http://grigoriefflab.janelia.org/ctf | |
| Software, algorithm | DINO 0.9.4 | http://www.dino3d.org | http://www.dino3d.org | |
| Software, algorithm | EPU 2.9 | Thermo Fisher Scientific | N/A | |
| Software, algorithm | PHENIX 1.14 | *Afonine et al., 2018* | http://phenix-online.org/ | |
| Software, algorithm | RELION-4.0 | *Scheres, 2012* | https://www2.mrc-lmb.cam.ac.uk/relion/ | |
| Software, algorithm | Prism 10 | GraphPad | https://www.graphpad.com/ | |
| Other | 200 mesh Au 1.2/1.3 cryo-EM grids | Quantifoil | N1-C14nAu20-01 | See Methods, Cryo-EM sample preparation, and data collection |
| Other | 0.22 µm Ultrafree-MCCentrifugal Filter | Millipore | UFC30GV | See Methods, CALHM protein purification |
| Other | Amicon 100 kDa MWCO centrifugal filter | Millipore | UFC810096 | See Methods, CALHM protein purification |
| Other | NiNTA agarose beads | ABT | 6BCL-NTANi-100 | See Methods, CALHM protein purification, Sybody expression, and purification |
| Other | PD-10 desalting column | Sigma | GE17-0851-01 | See Methods, Sybody selection |
| Other | SRT-10C SEC 100 | Sepax Technologies | 239100–10,030 | See Methods, Sybody expression and purification |
| Other | Strep-Tactin Superflow high capacity 50% suspension | Lucerna-Chem (IBA) | 2-1208-010 | See Methods, CALHM protein purification, Pulldown binding assays |
| Other | Superose 6 10/300 GL | GE Healthcare | 17-5172-01 | See Methods, CALHM protein purification |

