## [Editor Report · eLife assessment]

In this interesting study, Drożdżyk and colleagues analyze the ability of placental CALHM orthologs to form stable complexes, identifying that CALHM2 and CALHM4 form heterooligomeric channels. The authors then determine cryo-EM structures of heterooligomeric CALHM2 and CALHM4 that reveal a distinct arrangement in which the two orthologs can interact, but preferentially segregate in the channel. This is an **important** study; the data provide **compelling** support for the interpretations and overall, the work is clearly described.

---

## [Referee Report · Reviewer #1 (Public review)]

The Calcium Homeostasis Modulators (CALHM) are a family of large pore channels, of which the physiological role of CALHM1 and 3 is well understood, in particular their key role in taste sensation via the release of the neurotransmitter ATP. The activation mechanism of CALHM1 involves membrane depolarization and a decrease in extracellular Ca concentration, allowing the passage of large cellular metabolites. However, the activation mechanism and physiological roles of other family members are much less well understood. Many structures of homomeric CALHM proteins have been determined, revealing distinct oligomeric assemblies despite a common transmembrane domain topology. CALHM1 and 3 have been shown functionally to form heteromeric assemblies with properties distinct from those of homomeric CALHM1. However, the structural basis of heteromeric CALHM1 and 3 remains unexplored.

In this paper, Drozdzyk et al. present an important study on the structures of heteromeric channels composed of CALHM2 and CALHM4, extending the structural understanding of the CALHM family beyond homomeric channels. The study relies primarily on cryo-EM. Despite the inherent challenges of structural determination due to the similar structural features of CALHM2 and CALHM4, the authors innovatively use synthetic nanobodies to distinguish between the subunits. Their results show a broad distribution of different heteromeric assemblies, with CALHM4 conformation similar to its homomeric form and CALHM2 conformation influenced by its proximity to CALHM4, and provide detailed insights into the interaction between CALHM2 and CALHM4.

The manuscript is well-structured and presents clear results that support the conclusions drawn. The discovery of heteromeric CALHM channels, although currently limited to an overexpressed system, represents a significant advance in the field of large-pore channels and will certainly encourage further investigation into the physiological relevance and roles of heteromeric CALHM channels.

Comments on the revised version:

I appreciate the authors' efforts to try the alternative data processing strategy. Congratulations to the authors for this interesting and important work!

---

## [Referee Report · Reviewer #2 (Public review)]

Summary:

The authors identified that two of the placental CALHM orthologs, CALHM2 and CALHM4 can form heterooligomeric channels that are stable following detergent solubilization. By adding fiducial markers that specifically recognize either CALHM2 or CALHM4, the authors determine a cryo-EM density map of heterooligomeric CALHM2/CALHM4 from which they can determine how the channel in assembled. Surprisingly, the two orthologs segregate into two distinct segments of the channel. This segregation enables the interfacial subunits to ease the transition between the preferred conformations of each ortholog, which are similar to the confirmation that each ortholog adopts in homooligomeric channels.

Strengths:

Through the use of fiducial markers, the authors can clearly distinguish between the CALHM2 and CALHM4 promoters in the heterooligomeric channels, strengthening their assignment of most of the promoters. The authors take appropriate caution in identifying two subunits that are likely a mix of the two orthologs in the channel.

Weaknesses:

Despite the authors' efforts, no currents could be observed that corresponded to CALHM2/CALHM4 channels and thus the functional effect of their interaction is not known.

---

## [Author Response]

The following is the authors’ response to the original reviews.

We thank both reviewers for their supportive comments. Reviewer 1 has suggested a different data processing strategy to better resolve subunits at the CALHM4/CALHM2 interface:

I recommend an alternative data processing strategy. First, refine particles with 2-4 CALHM4 subunits with symmetry imposed. This is followed by symmetry expansion, signal subtraction of two adjacent subunits, and subsequent classification and refinement of the subtracted particles. This approach, while not guaranteed, can potentially provide a clearer definition of CALHM2 and CALHM4 interfaces and show whether CALHM2 subunits adopt different conformations based on their proximity to CALHM4 subunits.

We have followed the recommended strategy in an attempt to improve the resolution and better resolve the structural heterogeneity in CALHM2/4 channels. To this end, we have combined symmetry expansion and partial signal subtraction, as suggested by the reviewer. Initially, a symmetrized (C11) 3.4 Å consensus map of undecameric CALHM2/4 channels bound to sybodies SbC2 and SbC4 was used. The particles of this reconstruction were subjected to symmetry expansion (C11) followed by signal subtraction of nine adjacent subunits. Next, we performed focused, alignment-free 3D classification of the remaining two subunits followed by refinement of these classes, leading to the classification of CALHM subunit pairs. The majority of the classes feature well-resolved CALHM2 pairs, consistent with the original approach (Author response image 1). A minority of the classes contain CALHM4 subunits, revealing heterogeneity similar to regions of CALHM4 subunits observed in the non-symmetrized channel reconstruction (Author response image 1) . Unfortunately, this approach thus did not improve resolution or facilitate a more accurate subunit assignment. Consequently, we decided not to include these attempts in our manuscript. The resubmitted version thus contains only small corrections compared to the previous version.

**Author response image 1. sa3fig1:** Classification of subunit pairs of undecameric CALHM2/4 channels bound to sybodies SbC2 and SbC4 after the processing combining symmetry expansion and partial signal subtraction. (A) Classes showing CALHM2 subunit pairs. (B) Classes showing subunits at interfaces to CALHM4.